# Flatness Guided Test-Time Adaptation for Vision-Language Models

**Aodi Li**[1], **Liansheng Zhuang**[1,2(✉)], **Xiao Long**[1], **Houqiang Li**[2] **& Shafei Wang**[3]

[1]School of Cyber Science and Technology, University of Science and Technology of China
[2]National Engineering Laboratory for Brain-Inspired Intelligence Technology and Applications,
 University of Science and Technology of China    [3]Peng Cheng Laboratory, Shenzhen, China
`aodili@mail.ustc.edu.cn, lszhuang@ustc.edu.cn`

## Abstract

Test-time adaptation (TTA) of Vision-Language Models (VLMs) has emerged as a technique for tackling distribution shifts during the test time. Recent research indicates that the test-time adaptation is intrinsically linked to the model's training history. However, existing TTA methods, such as Test-time Prompt Tuning, often design adaptation strategies in isolation from the models' training characteristics, which degrade their performance. This paper argues that the flatness acquired via sharpness-aware training is an efficient clue for the test-time adaptation of VLMs. Built on this insight, this paper proposes a novel Flatness-Guided Adaptation framework (FGA) for VLMs to cohesively unify training and test-time procedures. Its core idea is to leverage the alignment between the training minimum and test loss flat regions to guide the adaptation process. Specifically, our FGA consists of a prompt-tuning stage and a test-time adaptation stage. In the tuning stage, a Sharpness-Aware Prompt Tuning method is utilized to identify the training flat minimum, offering a geometric clue of flatness for subsequent adaptation. In the test stage, a Sharpness-based Test Sample Selection approach is proposed to ensure the alignment of flat minima between the training and each augmented test sample's loss landscape. In comparison to existing TTA methods, our FGA avoids the expensive prompt parameter updates during test time, and substantially reduces the computation overhead. Extensive experiments on both domain generalization and cross-dataset benchmarks demonstrate that our FGA achieves superior performance over prevalent TTA methods. Notably, when employing a ViT-B/16 image encoder, FGA even outperforms TPT+CoOp by an average of 4.88% across all four ImageNet out-of-domain variants.

## 1 Introduction

Recent advancements in vision-language pretraining, such as CLIP (Radford et al., 2021), have generated new opportunities for developing foundational models in vision tasks (Jia et al., 2021; Yang et al., 2022). These models, trained on extensive collections of image-text pairs, can learn and represent a diverse range of visual concepts. By means of well-designed prompts, they can be applied to downstream tasks in a zero-shot manner without requiring task-specific data (Li et al., 2022; Ramesh et al., 2022; Patashnik et al., 2021). Consequently, various prompt tuning methods (Zhou et al., 2022b;a) are proposed to directly learn prompts using training data from downstream tasks. Though these methods find better prompts compared to hand-crafted ones, the learned prompts are limited to the training distribution and may have limited generalization beyond that.

To address this issue, several studies (Chen et al., 2022; Boudiaf et al., 2022; Wang et al., 2020) have attempted to develop test-time adaptation (TTA) methods, which aim to rapidly adjust pre-trained models to unlabeled test data streams during inference. Among various TTA strategies, methods based on Test-Time Prompt Tuning (TPT) (Shu et al., 2022), which optimize a set of learnable prompts via entropy minimization on augmented test views, have demonstrated promising performance and gained significant attraction (Shu et al., 2022; Feng et al., 2023; Yoon et al., 2024). Recent research indicates that the test-time adaptation is intrinsically influenced by the model's training history (Goyal et al., 2022). However, most existing TTA methods, including TPT-based methods,

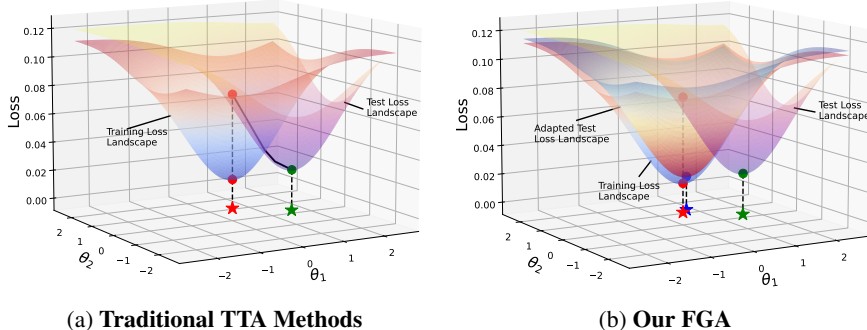

(a) **Traditional TTA Methods**      (b) **Our FGA**

Figure 1: **Comparison of conventional TTA methods and Flatness-Guided Adaptation (FGA).** (a) Traditional TTA methods treat the test landscape as static, aiming to optimize parameters to achieve the test flat minimum (★), using the training minimum (★) as an initialization. (b) Our FGA keeps parameters unchanged during testing. Instead, it adjusts the test landscape to a position where the training minimum (★) is already very close to the minimum (★) of the adapted test landscape.

design adaptation strategies in isolation, treating the test phase as a standalone optimization problem disconnected from the model's training history. This isolation from the training phase may fail to exploit valuable geometric and representational properties inherent in the pre-trained model, leading to suboptimal test-time adaptation.

To improve model generalization, seeking flat minima within the training loss landscape has emerged as an effective training strategy over the past few years (Foret et al., 2020; Kwon et al., 2021; Kim et al., 2022). It is widely observed that parameters residing in flat minima tend to generalize better to out-of-distribution data (Cha et al., 2021; Zhang et al., 2024b; Li et al., 2025; Zou et al., 2024) than those sharp ones. Nonetheless, conventional TTA methods often ignore the influence of sharpness-aware training on the test-time adaptation. While Sharpness-Aware Minimization (SAM) (Foret et al., 2020) seeks flat regions during training, its principle is rarely extended to guide test-time adaptation in a unified framework. This disconnection leads to computationally expensive test-time optimizations (e.g., backpropagation in TPT (Shu et al., 2022)) that are agnostic to loss geometric structure and often yield suboptimal generalization. This paper argues that flatness is not merely a desirable property during training but a powerful clue that can dictate test-time adaptation.

Inspired by this insight, this paper proposes a novel *Flatness-Guided Adaptation* (FGA) framework, which cohesively unifies training and test-time procedures from the perspective of loss landscape geometry. It mainly leverages the alignment between the training minimum and test loss flat regions to guide the adaptation process (see Figure 1). Specifically, our FGA framework consists of two synergistic stages: (1) In the prompt tuning stage, a *Sharpness-Aware Prompt Tuning* (SAPT) method is utilized to fine-tune the prompts on the downstream training dataset, aiming at seeking the training flat minimum. Since flatter minima generally indicate better model generalization than sharper ones (Keskar et al., 2016; Dziugaite & Roy, 2017; Jiang et al., 2019; Foret et al., 2020), the minimization of sharpness not only improves model generalization but also provides a test-time criterion to measure the alignment of flat minima within the training and test loss landscapes. (2) In the test-time stage, FGA leverages the geometric clue of flatness acquired via SAPT. For a given test sample, a *Sharpness-based Test Sample Selection* (STSS) method is proposed to intelligently select its augmented views based on the sharpness score of their loss landscapes around the training flat minimum. This ensures that the final prediction is derived from a test-time loss landscape whose flat minima align with those identified during training. During this process, loss landscapes are efficiently altered through data augmentations. In comparison with existing TTA methods, our FGA avoids the expensive prompt parameter updates during test time, eliminating the computational overhead of adaptation and offering a more plausible adaptation strategy. Theoretical analysis suggests that using the sharpness-based metric will help distinguish the proximity of test samples to the training distribution. The closer an augmented sample is to the training distribution, the smaller its sharpness-based score is likely to be. Since models tend to generate more reliable results for data closer to the training distribution, FGA significantly improves the generalization ability of vision-language models. Extensive experiments on domain generalization (Hendrycks et al., 2021b) and cross-dataset (Zhou et al., 2022a) benchmarks demonstrate the superior performance of FGA over prevailing TTA methods.

Our main contributions can be summarized as follows:

- A novel Flatness-Guided Adaptation (FGA) framework is proposed to cohesively unify training and test-time procedures for vision-language models. By ensuring the alignment of the model's training flat minimum with flat regions in test loss landscapes, it significantly enhances the generalization capabilities of VLMs under distribution shifts.

- Theoretical analysis is presented to offer a clearer insight into how sample selection at test time improves the reliability of predictions.

- Extensive experiments on domain generalization and cross-dataset benchmarks demonstrate the superior performance of FGA over other prevalent TTA methods, while significantly eliminating the computational overhead.

## 2 RELATED WORK

**Test-time adaptation (TTA) of vision-language models.** Vision-language models like CLIP have shown strong performance in various tasks. To enhance CLIP's transfer learning for downstream classification tasks, methods like text prompt learners (e.g., CoOp (Zhou et al., 2022b) and Co-CoOp (Zhou et al., 2022a)) and visual adapters (e.g., Tip-Adapter (Zhang et al., 2022)) have been proposed. However, these methods struggle with distribution misalignment between pre-training and test data. Test-time adaptation (TTA) methods address this by adjusting models during testing, with two main streams (Abdul Samadh et al., 2024): the first modifies the training process using a self-supervised proxy task, such as image rotation prediction, and uses it to guide test-time optimization (e.g., Test-Time Training (Sun et al., 2020) and TTT++ (Liu et al., 2021)); the second adapts models without altering the training process (e.g., TPT (Shu et al., 2022), which uses entropy minimization to learn adaptive parameters during testing). DiffTPT (Feng et al., 2023) introduces a diffusion model to generate diverse augmentations for further improvements. PromptAlign (Abdul Samadh et al., 2024) adds an explicit term to align the learned distributions with that of test data. Meanwhile, online methods like TDA (Karmanov et al., 2024) and DPE (Zhang et al., 2024a) use a key-value cache or prototype set to adapt progressively to test data. They benefit from information aggregated during testing but are unsuitable for single test-sample scenarios, unlike TPT-based methods. This paper proposes a novel Flatness-Guided Adaptation (FGA) framework that leverages the geometry of loss landscapes to enhance CLIP's generalization and inference efficiency in single test-sample adaptation scenarios. By avoiding backpropagation and parameter updates during testing, FGA significantly reduces computational overhead while achieving robust out-of-domain performance.

**Generalization from a loss landscape view.** In recent years, optimization techniques aimed at flat minima in loss landscapes have surged to improve the generalization of deep models (Keskar et al., 2016; Dziugaite & Roy, 2017; Jiang et al., 2019). Among them, SAM (Foret et al., 2020), which focuses on finding parameters located in regions of the loss landscape with consistently low loss values, has gained significant attention for its effectiveness and scalability. To seek flatter minima, numerous SAM variants, such as ASAM (Kwon et al., 2021) and FisherSAM (Kim et al., 2022), have already been developed over the past few years. This concept of flat minima has also been extended to improve the out-of-domain generalization of deep models (Zou et al., 2024; Cha et al., 2021; Li et al., 2025). However, most of them focus on the training stage. SAR (Niu et al., 2023) and SoTTA (Gong et al., 2023), two online test-time adaptation (TTA) methods, both utilize sharpness-aware minimization at test time to improve robustness by seeking flat minima. Yet, they operate solely during testing without accounting for training-testing sharpness interactions. In contrast, our FGA applies sharpness-aware minimization during training to establish flatness as a criterion for subsequent alignment, then at test time adapts by adjusting loss landscapes through augmentation selection (without updating model parameters), and preserving the pre-trained flat minimum's optimality on adapted test loss landscapes.

## 3 METHODOLOGY

### 3.1 PRELIMINARIES

**Contrastive Language-Image Pre-training.** CLIP (Radford et al., 2021) primarily comprises a Text Encoder $E_t$ and an Image Encoder $E_v$. The Image Encoder is available in two architectures:

one based on ResNet (He et al., 2016) and the other using the popular Vision Transformer (ViT) (Dosovitskiy et al., 2020). This encoder transforms an input image $x$ into its feature representation, i.e., $e_i = E_v(x)$. For a classification task with $K$ classes, the $k$-th class label is formatted into the textual prompt "a photo of a [$k$-th class]", which is then tokenized into a sequence $l_k = (t_{SOS}, t_1, t_2, \ldots, t_L, c_k, t_{EOS})$. Here, $t_{SOS}$ and $t_{EOS}$ represent the embeddings of the start and end tokens, while $t_1, t_2, \ldots, t_L$ corresponds to the phrase "a photo of a", and the token $c_k$ denotes the specific description of the $k$-th class. The text encoder of CLIP, designed as a Transformer architecture, processes these tokens to generate text features: $e_{t,k} = E_t(l_k)$. During the pre-training stage, CLIP is trained on the WIT dataset (Radford et al., 2021) through a contrastive learning approach. In this setup, each image is paired with its corresponding text sentence as a positive sample, while all other image-text combinations are treated as negative samples. The goal of the contrastive learning objective is to enhance the cosine similarity of positive pairs while reducing that of negative pairs. In the classification stage, all classes in the dataset are converted to text, and the cosine similarity between image embeddings and text embeddings is computed to determine the probability of an image belonging to each category:

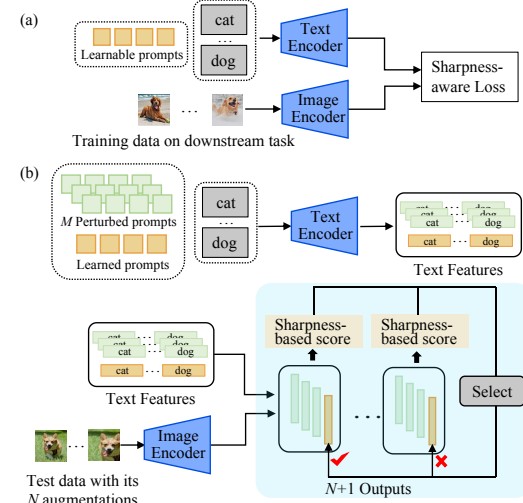

Figure 2: **Overview of our Flatness-Guided Adaptation (FGA).** It consists of two synergistic mechanisms: (a) Sharpness-aware Prompt Tuning: It optimizes the model parameters to reduce the loss value and sharpness, enabling stable and effective adaptations during test time without direct access to training data. (b) Sharpness-based Test Sample Selection: It introduces a selection mechanism to identify augmented test samples that ensure the training flat minimum aligns with those in their loss landscapes, enabling more confident predictions.

$$\mathbb{P}\left(y_k \mid x\right) = \frac{\exp\left(\text{sim}\left(e_{t,k} \cdot e_i\right)\tau\right)}{\sum_{j=1}^{K} \exp\left(\text{sim}\left(e_{t,j} \cdot e_i\right)\tau\right)}, \tag{1}$$

where $\tau$ is the temperature of the softmax.

**Prompt tuning.** Prompt tuning has emerged as a popular tuning method for Transformer-based models in downstream tasks. This approach does not modify the model parameters; rather, it changes the input to the model, making it highly efficient. Specifically, instead of using the template "a photo of a [class]", it replaces the tokens associated with the hand-crafted prompts ("a photo of a") with learnable parameters $p = (p_1, \ldots, p_L)$, which are then updated based on the dataset used for downstream tasks.

**Test-time prompt tuning.** Prompts optimized on downstream training data are prone to overfitting and often generalize poorly under distribution shifts. To mitigate this, Test-Time Prompt Tuning (TPT) (Shu et al., 2022) fine-tunes a specific prompt for each individual test sample. During testing, multiple augmented views of the test samples are generated. Then, predictions with entropy below a predetermined threshold are kept, while others are discarded using a confidence filter. The averaged entropy of selected predictions is then used as a loss function to update the prompts.

## 3.2 FLATNESS-GUIDED ADAPTATION

Our proposed Flatness-Guided Adaptation (FGA) framework fundamentally offers a unified, loss landscape-centric methodology that seamlessly bridges the training and test phases. This approach leverages the flatness as a universal guiding principle to enhance both generalization during training and robust adaptation during inference under distribution shifts. As illustrated in Figure 2, FGA integrates two complementary mechanisms that operate in concert: sharpness-aware prompt tuning (SAPT) during training and sharpness-guided test sample selection (STSS) during inference. This section will elaborate on the key idea and technical details of the framework.

### 3.2.1 SHARPNESS-AWARE PROMPT TUNING

FGA mainly exploits the alignment between training and test flat minima for the efficient adaptation of VLMs. However, a major challenge in achieving this alignment arises from inherent data constraints. Test samples remain unknown during training, and at test time, training data becomes inaccessible due to storage limitations and privacy requirements. To overcome this, FGA focuses on the intrinsic properties of the training minimum, with the goal that the test loss landscape exhibits these same desirable characteristics: (1) The training minimum should correspond to a low loss value, indicating that the model is effectively learning from the data; (2) The training minimum may display implicit biases, such as reduced sharpness, which are often beneficial for improved generalization.

Traditional training methods for optimizing the prompts, such as CoOp (Zhou et al., 2022b), typically use cross-entropy loss to fine-tune the prompt $\boldsymbol{p}$:

$$\ell_{\mathrm{CE}}(\boldsymbol{p}) = -\sum_{i=1}^{n} \log \mathbb{P}_{\boldsymbol{p}}(y_i|\boldsymbol{x}_i), \tag{2}$$

where $p_{\boldsymbol{p}}(y_i|\boldsymbol{x}_i)$ represents the predictive probability that $\boldsymbol{x}_i$ belongs to the its true label class. While standard SGD tends to find flat minima, methods such as SAM (Foret et al., 2020) can enhance this implicit bias through explicit perturbation-aware optimization. To enable more precise alignment during testing, we adopt Sharpness-aware Prompt Tuning (SAPT) during training, which jointly minimizes both the loss and its "sharpness":

$$\ell_{\mathrm{SAPT}}(\boldsymbol{p}) = \ell_{\mathrm{CE}}(\boldsymbol{p}) + \lambda \max_{||\boldsymbol{\epsilon}|| \leq \rho} \left[ \ell_{\mathrm{CE}}(\boldsymbol{p} + \boldsymbol{\epsilon}) - \ell_{\mathrm{CE}}(\boldsymbol{p}) \right]. \tag{3}$$

The first and second terms above represent the loss value and loss sharpness, with $\lambda$ acting as a hyperparameter to balance them. Similar to previous studies (Foret et al., 2020; Kwon et al., 2021), sharpness is defined as the sensitivity of the training loss to small perturbations $\boldsymbol{\epsilon}$ (with a norm less than $\rho$) added to the prompts $\boldsymbol{p}$. Since the perturbation strength $\rho$ is small enough, we can apply a Taylor expansion to approximately solve for the optimal perturbation $\boldsymbol{\epsilon}^\star$:

$$\boldsymbol{\epsilon}^\star = \arg\max_{||\boldsymbol{\epsilon}|| \leq \rho} \ell_{\mathrm{CE}}(\boldsymbol{p} + \boldsymbol{\epsilon}) - \ell_{\mathrm{CE}}(\boldsymbol{p}) \approx \arg\max_{||\boldsymbol{\epsilon}|| \leq \rho} \boldsymbol{\epsilon}^\top \nabla_{\boldsymbol{p}} \ell_{\mathrm{CE}}(\boldsymbol{p}) = \rho \frac{\nabla_{\boldsymbol{p}} \ell_{\mathrm{CE}}(\boldsymbol{p})}{\|\nabla_{\boldsymbol{p}} \ell_{\mathrm{CE}}(\boldsymbol{p})\|}. \tag{4}$$

During training via stochastic gradient descent, the contribution from $\nabla_{\boldsymbol{p}} \boldsymbol{\epsilon}^\star$ can be disregarded due to the minor perturbation strength $\rho$.

In this way, SAPT not only yields robust prompts that enhance generalization but also provides the sharpness measure as additional information for adaptation during testing.

### 3.2.2 SHARPNESS-BASED TEST SAMPLE SELECTION

Through sharpness-aware prompt tuning, the prompts are positioned at a flat minimum within the training loss landscape. To avoid computationally expensive gradient descent during inference, we keep the pre-trained prompt fixed and instead adapt the test loss landscapes such that the well-trained prompt from the downstream training dataset (i.e., the training flat minimum) coincides with the flat minimum in the adapted test landscape, as illustrated in Figure 1.

To achieve this alignment, we propose a Sharpness-based Test Sample Selection (STSS) method. STSS utilizes data augmentations to create multiple test loss landscapes for each sample. By selecting augmented samples that align the training minimum with flat minima in their respective loss landscapes, we ensure the training minimum remain optimal. Given that such alignment typically corresponds to small loss values and reduced loss sharpness in these test landscapes, STSS introduces a sharpness-based score as a metric. To mitigate the computational burden of backpropagation in calculating sharpness, we redefine it as the maximum variation in the loss resulting from $R$ random perturbations:

$$\ell_{\mathrm{STSS}}(\boldsymbol{p}) = \ell_{\mathrm{SRG}}(\boldsymbol{p}) + \lambda \max_{r=1,\ldots,R; \boldsymbol{\epsilon}_r \sim \mathcal{N}} \left[ \ell_{\mathrm{SRG}}\left(\boldsymbol{p} + \rho' \frac{\boldsymbol{\epsilon}_r}{\|\boldsymbol{\epsilon}_r\|}\right) - \ell_{\mathrm{SRG}}(\boldsymbol{p}) \right]. \tag{5}$$

Here, $\ell_{\mathrm{STSS}}$ represents the sharpness-based score used to select the most reliable augmented test samples, and $\ell_{\mathrm{SRG}}$ denotes a surrogate loss function when test labels are unavailable, such as entropy (Wang et al., 2020; Goyal et al., 2022). The perturbation direction is expressed by $\boldsymbol{\epsilon}_r/\|\boldsymbol{\epsilon}_r\|$,

where $\epsilon_r$ is drawn from the standard normal distribution $\mathcal{N}$. The quantity $\rho'$ controls the magnitude of perturbations during testing. To obtain $\ell_{\mathrm{SRG}}\left(\boldsymbol{p} + \rho' \frac{\epsilon_r}{\|\epsilon_r\|}\right)$, we first obtain text features for each category ($\boldsymbol{e}_{t,k,r}$) through the forward pass of the text encoder:

$$[\boldsymbol{e}_{t,k,1}, \ldots, \boldsymbol{e}_{t,k,R}] = \boldsymbol{E}_t([\boldsymbol{l}_{k,1}, \ldots, \boldsymbol{l}_{k,R}]), \tag{6}$$

where $\boldsymbol{l}_{k,r}$ denotes the input token related to the $k$-th category with the $r$-th pertuation. Notably, the additional computational cost of this step is minimal, as text features only need to be computed once per test category. Then, the surrogate loss for perturbed prompts is:

$$\ell_{\mathrm{SRG}}\left(\boldsymbol{p} + \rho' \frac{\epsilon_r}{\|\epsilon_r\|}\right) = -\sum_{k=1}^{K} \mathbb{P}_r(y_k|\boldsymbol{x}) \log \mathbb{P}_r(y_k|\boldsymbol{x}), \tag{7}$$

with probabilities derived from cosine similarity:

$$\mathbb{P}_r(y_k|\boldsymbol{x}) = \frac{\exp\left(\mathrm{sim}\left(\boldsymbol{e}_{t,k,r} \cdot \boldsymbol{e}_i\right)\tau\right)}{\sum_{j=1}^{K} \exp\left(\mathrm{sim}\left(\boldsymbol{e}_{t,j,r} \cdot \boldsymbol{e}_i\right)\tau\right)}. \tag{8}$$

Finally, the final prediction aggregates votes from the top $s$ augmented samples with the lowest sharpness-based scores, which are more reliable predictions according to the theoretical analysis in the next section.

## 4 THEORETICAL ANALYSIS

This section provides a theoretical explanation of how our method improves test-time classification. Let's begin with the following problem: During training, the model learns from data sampled independently and identically from distribution $\mathcal{S}$; During testing, however, data is drawn from two distinct distributions $\mathcal{T}_1$ and $\mathcal{T}_2$. Then, the question is: *How can we distinguish between these test distributions and determine on which one the model will perform more reliably?*

To address this, we first derive an upper bound for the generalization error, which quantifies the model's performance on unseen data from $\mathcal{T}_1$ and $\mathcal{T}_2$. We will then explore how, when the test distributions are sufficiently distinguishable, FGA can effectively distinguish between them. This is crucial because, as we will show, when the test distribution closely resembles the training distribution, the generalization error bound decreases, leading to more accurate predictions.

**Theorem 1 (Generalization Bound)** *Consider a vector-valued function class $\mathcal{F}_v = \{\boldsymbol{f_\theta}(\cdot) : \mathcal{X} \to \mathbb{R}^K\}$ and a bounded loss function $\ell : \mathbb{R}^K \times \mathcal{Y} \to [0, M]$. Define the worst-case loss $\ell^\rho$ within a $\rho$-radius perturbation as*

$$\ell^\rho(\boldsymbol{f_\theta}(\boldsymbol{x}), y) = \max_{\|\epsilon\|_2 \leq \rho} \ell(\boldsymbol{f_{\theta+\epsilon}}(\boldsymbol{x}), y). \tag{9}$$

*Assume $\ell^\rho$ is $\mu$-Lipschitz continuous with respect to $\boldsymbol{f}$. Let $\mathcal{S}$ and $\mathcal{T}$ denote the training and test distributions, respectively. Denote $\mathcal{S}_n = \{\boldsymbol{x}_i, y_i\}_{i=1}^n$ as $n$ i.i.d. training samples drawn from distribution $\mathcal{S}$. Then, with probability at least $1 - \delta$, the following generalization bound holds:*

$$\mathbb{E}_{\mathcal{T}}\left[\ell^\rho(\boldsymbol{f_\theta}(\boldsymbol{x}_\tau), y_\tau)\right] \leq \frac{M}{2} d(\mathcal{S}; \mathcal{T}) + \hat{\ell}_{\mathcal{S}_n}^\rho(\boldsymbol{f_\theta}) + 2\sqrt{2}\,\mu\, R_n(\mathcal{F}, \mathcal{S}) + M\sqrt{\frac{\log(1/\delta)}{2n}}, \tag{10}$$

*where $(\boldsymbol{x}_\tau, y_\tau)$ is a random pair following the distribution $\mathcal{T}$; $\hat{\ell}_{\mathcal{S}_n}^\rho(\boldsymbol{f_\theta}) := \frac{1}{n}\sum_{i=1}^n \ell^\rho(\boldsymbol{f_\theta}(\boldsymbol{x}_i), y_i)$ denotes the empirical loss over $\mathcal{S}_n$; $d(\mathcal{S}; \mathcal{T})$ is the distribution discrepancy measure defined in Appendix A; and $R_n(\mathcal{F}_v, \mathcal{S})$ represents the Rademacher complexity of $\mathcal{F}_v$ on $\mathcal{S}$.*

In the following, we will show that when the two test distributions are sufficiently distinguishable—compared with the tightness of the above upper bound—we can effectively differentiate between them. To proceed with this analysis, we first introduce the concepts of bound tightness and distribution separability.

**Definition 2 ($\beta$-tightness)** *Given a random variable $X$ and a confidence level $\delta \in (0, 1)$, let $\alpha$ be an upper bound such that $\mathbb{P}(X \leq \alpha) \geq 1 - \delta$. If there exists an oracle upper bound $\alpha^\star$ satisfying $\mathbb{P}(X \leq \alpha^\star) = 1 - \delta$, then $\alpha$ is said to be $\beta$-tight with $\beta = |\alpha - \alpha^\star|$.*

Table 1: **Results on datasets with natural distribution shifts.** We report top-1 accuracy (%) for each method across five datasets, using the CLIP-ViT-B/16 backbone. We highlight the best results in **bold** and underline the second-best results. The abbreviation "IN" means the ImageNet dataset. "†" denotes results reproduced by adapting the method to the single-sample setting.

| Algorithm | IN | IN-A | IN-V2 | IN-R | IN-Sketch | Avg. | OOD Avg. |
|---|---|---|---|---|---|---|---|
| CLIP-ViT-B/16 (Radford et al., 2021) | 68.34 | 49.89 | 61.88 | 77.65 | 48.24 | 61.20 | 59.42 |
| CoOp (Zhou et al., 2022b) | 71.51 | 49.71 | 64.20 | 75.21 | 47.99 | 61.72 | 59.28 |
| CoCoOp (Zhou et al., 2022a) | 71.02 | 50.63 | 64.07 | 76.18 | 48.75 | 62.13 | 59.91 |
| Tip-Adapter (Zhang et al., 2022) | 70.75 | 51.04 | 63.41 | 77.76 | 48.88 | 62.37 | 60.27 |
| TPT (Shu et al., 2022) | 69.70 | 53.67 | 64.30 | 73.90 | 46.40 | 61.59 | 59.57 |
| DiffTPT (Feng et al., 2023) | 70.30 | 55.68 | 65.10 | 75.00 | 46.80 | 62.58 | 60.64 |
| C-TPT (Yoon et al., 2024) | - | 52.90 | 63.40 | 78.00 | 48.50 | - | 60.70 |
| ZERO (Farina et al., 2024) | 69.06 | 61.35 | 64.13 | 77.28 | 48.29 | 64.02 | 62.76 |
| MTA (Zanella & Ben Ayed, 2024) | 69.29 | 57.41 | 63.61 | 76.92 | 48.58 | 63.16 | 61.63 |
| PromptAlign* (Abdul Samadh et al., 2024) | - | 59.37 | 65.29 | 79.33 | 50.23 | - | 63.56 |
| TDA (Karmanov et al., 2024) | 69.51 | 60.11 | 64.67 | 80.24 | 50.54 | 65.01 | 63.89 |
| DPE (Zhang et al., 2024a) | 71.91 | 59.63 | 65.44 | 80.40 | **52.26** | 65.93 | 64.43 |
| TPT (Shu et al., 2022)+CoOp | 73.30 | 56.88 | 66.60 | 73.80 | 49.40 | 64.00 | 61.67 |
| DiffTPT (Feng et al., 2023)+CoOp | **75.00** | 58.09 | 66.80 | 73.90 | 49.50 | 64.66 | 62.07 |
| C-TPT (Yoon et al., 2024)+CoOp | 72.90 | 52.73 | 65.61 | 76.46 | 48.63 | 63.27 | 60.86 |
| ZERO (Farina et al., 2024)+CoOp | 73.61 | 63.17 | 66.82 | 77.71 | 48.52 | 65.97 | 64.06 |
| MTA (Zanella & Ben Ayed, 2024)+CoOp | 73.99 | 59.29 | 66.97 | 78.20 | 49.96 | 65.68 | 63.61 |
| SAR† (Niu et al., 2023)+CoOp | 73.03 | 55.35 | 65.89 | 77.09 | 48.65 | 64.00 | 61.75 |
| **SAPT+CoOp** | 70.79 | 51.04 | 64.41 | 77.66 | 49.31 | 62.64 | 60.61 |
| **STSS+CoOp** | 73.99 | 64.00 | 67.11 | 77.92 | 49.36 | 66.48 | 64.60 |
| **FGA (Ours)** | 74.01 | **65.90** | **67.23** | **81.24** | 51.81 | **68.04** | **66.55** |

**Definition 3 ($\gamma$-separability)** *Let $\mathcal{T}_1$ and $\mathcal{T}_2$ be two test distributions and $\mathcal{S}$ the training distribution. Denote by $d(\cdot; \mathcal{S})$ the discrepancy between a distribution and $\mathcal{S}$. Then $\mathcal{T}_1$ and $\mathcal{T}_2$ are called $\gamma$-separable if $\left| d(\mathcal{T}_1; \mathcal{S}) - d(\mathcal{T}_2; \mathcal{S}) \right| > \gamma$.*

**Theorem 4** *Consider a vector-valued function class $\mathcal{F}_v = \{ \boldsymbol{f}_{\boldsymbol{\theta}}(\cdot) : \mathcal{X} \to \mathbb{R}^K \}$, where the parameters $\boldsymbol{\theta}$ lie in a set $\Theta$ such that the loss function is bounded within $[0, M]$. Let $\boldsymbol{f} = (f_1, \ldots, f_K)$ and $\mathbf{y} = (\mathrm{y}_1, \ldots, \mathrm{y}_K)$ be probability distributions over a finite set $\{1, \ldots, K\}$, with $f_i, \mathrm{y}_i \geq \eta > 0$ for all $i$. Denote by $H(\boldsymbol{f})$ the entropy and by $H(\mathbf{y}, \boldsymbol{f})$ the cross entropy. Assume the worst-case cross entropy $H^\rho(\mathbf{y}, \boldsymbol{f})$ is $\mu$-Lipschitz continuous with respect to $\boldsymbol{f}$ over $\Theta$. Given a training distribution $\mathcal{S}$ and two $\gamma$-separable test distributions $\mathcal{T}_1$ and $\mathcal{T}_2$, assume $d(\mathcal{S}, \mathcal{T}_1) < d(\mathcal{S}, \mathcal{T}_2)$. Define the quantile function $Q_i(\delta)$ for the entropy loss of $\boldsymbol{f}_{\boldsymbol{\theta}}$ on $\mathcal{T}_i$ such that $\mathbb{P}\left( H^\rho < \mathbb{E}[H^\rho] + Q_i(\delta) \right) = 1 - \delta$, and let $Q(\delta) = \sup\{Q_1(\delta), Q_2(\delta)\}$ be the supremum quantile. Then, with probability at least $1 - \delta$:*

$$H^\rho(\boldsymbol{f}_{\boldsymbol{\theta}}(\boldsymbol{x}_{\mathcal{T}_i})) \leq \frac{M}{2} d\left(\mathcal{S}; \mathcal{T}_i\right) + Q(\delta/2) + \hat{H}^\rho_{\mathcal{S}_n} + 2\mu R_n(\mathcal{F}_v, \mathcal{S}) + M\sqrt{\frac{\log(2/\delta)}{2n}}$$
$$+ \log \frac{1}{\eta} \cdot \mathbb{E}_{\mathcal{S}} \|\mathbf{y}_{\mathcal{S}} - \boldsymbol{f}_{\widetilde{\boldsymbol{\theta}}}(\boldsymbol{x}_s)\|_1 + \frac{1}{\eta} \mathbb{E}_{\mathcal{S}} \|\mathbf{y}_{\mathcal{S}} - \boldsymbol{f}_{\widetilde{\boldsymbol{\theta}}}(\boldsymbol{x}_s)\|_1^2. \tag{11}$$

*Here, the notation $\widetilde{\boldsymbol{\theta}}$ is defined as $\widetilde{\boldsymbol{\theta}} := \boldsymbol{\theta} + \arg\max_{\|\boldsymbol{\epsilon}\| \leq \rho} \max \{H(\mathbf{y}, \boldsymbol{f}_{\boldsymbol{\theta}+\boldsymbol{\epsilon}}(\boldsymbol{x})), H(\boldsymbol{f}_{\boldsymbol{\theta}+\boldsymbol{\epsilon}}(\boldsymbol{x}))\}$. Furthermore, if this bound is $\beta_i$-tight for $\mathcal{T}_1$ and $\mathcal{T}_2$ with $\beta_i < \gamma$, then there exists a threshold $\xi$ such that:*

$$\mathbb{P}\left( H^\rho(\boldsymbol{f}_{\boldsymbol{\theta}}(\boldsymbol{x}_{\mathcal{T}_1})) < \xi \right) > \mathbb{P}\left( H^\rho(\boldsymbol{f}_{\boldsymbol{\theta}}(\boldsymbol{x}_{\mathcal{T}_2})) < \xi \right). \tag{12}$$

This inequality indicates that a test distribution further from the training distribution tends to exhibit a higher sharpness score. By comparing sharpness scores across test distributions, we can identify which one is closer to the training distribution, thus yielding more reliable predictions. Notably, the tunable parameter $\rho$ controls the tightness of the upper bound, facilitating a precise differentiation between test distributions and improving the model performance. It is important to note that in the theoretical analysis presented in this section, we do not distinguish between $\rho$ and $\rho'$ (which are utilized to calculate the sharpness of the training and test loss landscapes, respectively). However, in practical implementation, we may opt to use different values for $\rho$ and $\rho'$ for better performance. Due to space limitations, detailed proofs and further discussions are provided in the appendices.

Table 2: **Cross-dataset generalization from ImageNet to fine-grained classification datasets.**
During the prompt tuning stage, the prompts are tuned on ImageNet with 16-shot training data per
category, using a ViT-B/16 image encoder.

| Method | Caltech101 | Pets | Cars | Flowers102 | Aircraft | SUN397 | DTD | Eurosat | Food101 | UCF101 | Avg. |
|---|---|---|---|---|---|---|---|---|---|---|---|
| CLIP-ViT-B/16 (Radford et al., 2021) | 93.35 | 88.25 | 65.48 | 67.44 | 23.67 | 62.59 | 44.27 | 42.01 | 83.65 | 65.13 | 63.58 |
| CoOp (Zhou et al., 2022b) | 93.70 | 89.14 | 64.51 | 68.71 | 18.47 | 64.15 | 41.92 | 46.39 | 85.30 | 66.55 | 63.88 |
| CoCoOp (Zhou et al., 2022a) | 94.43 | 90.14 | 65.32 | 71.88 | 22.94 | 67.36 | 45.73 | 39.23 | 83.97 | 68.44 | 64.94 |
| TPT (Shu et al., 2022) | 94.16 | 87.79 | 66.87 | 68.98 | 24.78 | 65.50 | _47.75_ | 42.44 | 84.67 | 68.04 | 65.10 |
| DiffTPT (Feng et al., 2023) | 92.49 | 88.22 | 67.01 | 70.10 | 25.60 | 65.74 | 47.00 | 43.13 | **87.23** | 62.67 | 64.92 |
| C-TPT (Yoon et al., 2024) | 93.60 | 88.20 | 65.80 | 69.80 | 24.00 | 64.80 | 46.00 | 43.20 | 83.70 | 65.70 | 64.48 |
| ZERO (Farina et al., 2024) | 93.66 | 87.75 | 68.04 | 67.68 | 25.21 | 65.03 | 46.12 | 34.33 | 86.53 | 67.77 | 64.21 |
| PromptAlign (Abdul Samadh et al., 2024) | 94.01 | _90.76_ | 68.50 | **72.39** | 24.80 | 67.54 | 47.24 | 47.86 | 86.65 | 69.47 | 66.92 |
| TDA (Karmanov et al., 2024) | 94.24 | 88.63 | 67.28 | 71.42 | 23.91 | 67.54 | 47.40 | **58.00** | 86.14 | **70.66** | _67.53_ |
| TPT+CoOp (Zhou et al., 2022b) | 93.75 | 88.93 | 67.06 | 68.25 | 25.89 | 66.40 | 47.15 | _48.78_ | 83.82 | 66.53 | 65.66 |
| TPT+MaPLe (Khattak et al., 2023) | 93.59 | 90.72 | 66.50 | _72.37_ | 24.70 | 67.54 | 45.87 | 47.80 | 86.64 | 69.19 | 66.50 |
| ZERO (Farina et al., 2024)+CoOp | 93.85 | 88.36 | 64.90 | 67.23 | 19.14 | 64.73 | 43.62 | 33.53 | 82.67 | 66.61 | 62.46 |
| ZERO (Farina et al., 2024)+MaPLe | _94.48_ | 90.60 | _68.58_ | 71.62 | _26.25_ | _68.20_ | 45.86 | 42.17 | _86.77_ | _69.87_ | 66.44 |
| **FGA (Ours)** | **96.96** | **91.28** | **68.93** | 72.11 | **26.97** | **69.29** | **49.76** | 47.58 | 84.95 | 68.17 | **67.60** |

## 5 EXPERIMENTS

### 5.1 EXPERIMENTAL SETUP

**Datasets.** We conduct two types of experiments to evaluate the model's robustness to natural distribution shifts and its cross-dataset generalization capabilities, following previous research such as TPT (Shu et al., 2022). To assess the model's robustness to natural distribution shifts, we apply prompt tuning on the ImageNet (Deng et al., 2009) dataset, and evaluate its performance on four ImageNet variants: ImageNet-A (Hendrycks et al., 2021c), ImageNet-V2 (Recht et al., 2019), ImageNet-R (Hendrycks et al., 2021a) and ImageNet-Sketch (Wang et al., 2019), which is also known as the domain generalization task. In addition, we perform cross-dataset evaluations for image classification across 10 datasets, each from a distinct domain with different classes: including Caltech101 (Fei-Fei et al., 2004), OxfordPets (Parkhi et al., 2012), StanfordCars (Krause et al., 2013), Flower102 (Nilsback & Zisserman, 2008), Aircraft (Maji et al., 2013), SUN397 (Xiao et al., 2010), DTD (Cimpoi et al., 2014), Food101 (Bossard et al., 2014), UCF101 (Soomro, 2012) and Eurosat (Helber et al., 2019). In this experiment, ImageNet serves as the source dataset, while the remaining fine-grained datasets are used as target datasets for evaluation.

**Implementation details.** Our experiments are based on pretrained CLIP (Radford et al., 2021) models, specifically CLIP-ResNet50 (using a ResNet50 image encoder) and CLIP-ViT-B/16 (using a Vision Transformer image encoder). Due to space limits, we focus on reporting the experimental results of CLIP-ViT-B/16, deferring those of CLIP-ResNet50 to the appendix. In the prompt tuning stage, our experiments are built on the CoOp (Zhou et al., 2022b) framework. The prompts are trained in a 16-shot manner on the ImageNet dataset. We set the number of prompts to 4 and utilize the SGD optimizer, with a learning rate of 0.002. For cross-dataset and domain generalization tasks, the prompts were trained for 5 and 50 epochs, with batch sizes of 4 and 32, respectively. The key hyperparameters $\rho$ are determined through a grid search, with the values ranging from [0.05, 0.1, 0.3, 0.5, 0.7]. During testing, existing TPT-based methods usually leverage the input image along with its 63 augmented views. To ensure a fair comparison, we consistently employ the same data augmentation strategy as TPT in all experiments. To avoid tuning hyperparameters on test data, we just set $\lambda = 1$ and $\rho' = 0.5$ for all experiments. Please refer to the Appendix for more discussions about other hyperparameters.

### 5.2 MAIN RESULTS

**Robustness to natural distribution shifts.** We first compare the proposed FGA with prevalent TTA techniques on ImageNet and its variant out-of-domain (OOD) datasets. The results, presented in Table 1, highlight the superior performance of FGA across several ImageNet-based OOD datasets. Notably, even the ablated version of our method, STSS+CoOp, exhibits strong performance, surpassing all previous approaches with an OOD average of 64.60% and an overall average of 66.48%. We attribute this robustness to the fact that standard SGD training already imbues models with an implicit bias toward flatter minima. By explicitly enhancing this geometric property through SAPT, the full

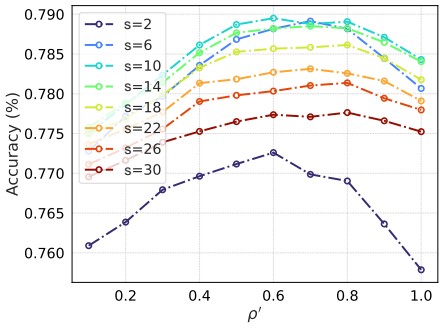

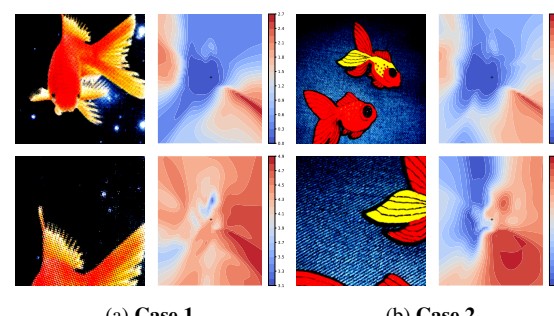

(a) **Case 1**       (b) **Case 2**

Figure 3: **The influence of the key hyper-parameter $\rho'$ on the test accuracy.**

Figure 4: **2D Visualization of loss landscapes associated with different augmented test samples.**

FGA algorithm achieves a substantial leap in generalization performance. Specifically, when compared to TPT+CoOp, FGA shows an average accuracy improvement of 4.88% (61.67% → 66.55%) on the OOD variants. Furthermore, our FGA also consistently surpasses other powerful TTA methods (e.g., DiffTPT, C-TPT, ZERO, MTA, and SAR) when they are combined with CoOp. It is critical to note that while online TTA methods like SAR benefit from aggregating information across a test data stream, FGA operates in a more challenging single-sample adaptation setting. For a fair comparison, we have adapted SAR to the single-sample adaptation setting. These superior results of FGA strongly demonstrate its effectiveness in enhancing CLIP's out-of-domain generalization across diverse datasets.

**Cross-dataset generalization.** We also observe superior performance of the FGA in evaluating cross-dataset generalization from ImageNet to various fine-grained classification benchmarks. Based on the comprehensive results presented in Table 2, the proposed FGA method demonstrates superior overall performance, achieving the highest average accuracy of 67.60% and attaining top-tier results on 6 out of 10 datasets, including a notably strong performance on Caltech101 (96.96%). Furthermore, FGA (67.60%) achieves an average accuracy improvement of 1.94% over the powerful baseline TPT+CoOp (65.66%). It also exhibits superior performance over the combinations of TTA methods (like TPT and ZERO) and different tuning methods (CoOp and MaPLe). Please note that due to the significant difference in TTA settings, it is not imperative to expect single-sample TTA methods to surpass online TTA methods like TDA. Overall, these superior results further validate its effectiveness in adapting to diverse datasets during testing. It is particularly valuable for VLMs like CLIP, as it enables models to recognize more fine-grained categories in image classification without the need for additional training.

**Runtime and memory efficiency.** To evaluate the practical deployment advantages of FGA, we quantify its computational overhead in terms of inference time per test image and peak GPU memory consumption. All experiments are conducted on a single NVIDIA Tesla V100 GPU under consistent settings. Specifically, FGA requires only 0.07s per image from ImageNet, which is 23.86× faster than DiffTPT (1.67s) and 8.86× faster than TPT (0.62s). In terms of memory usage, FGA consumes merely 4.14 GB, which is 4.67× lower than that of TPT (19.33GB). These results demonstrate that FGA achieves highly competitive test-time adaptation performance while maintaining notably low runtime and memory overhead, making it suitable for real-time and resource-constrained applications.

## 5.3 ABLATION STUDY

**Main components analysis.** Our ablation study on the domain generalization benchmark (using CLIP-ViT-B/16 architecture, shown in Table 1) validates the necessity of each FGA component: (1) Sharpness-aware prompt tuning (SAPT) enhances generalization, boosting CoOp's average accuracy by 0.92% (61.72%→62.64%) on ImageNet and OOD datasets; (2) Test-time sharpness selection (STSS) drives major gains, with CoOp+STSS outperforming CoOp by 4.76% (61.72%→66.48%); (3) SAPT synergistically enhances STSS, where full FGA (CoOp+SAPT+STSS) achieves a 5.40% gain over CoOp+SAPT (62.64%→68.04%) (exceeding standalone STSS improvements of 4.76%). This confirms that flatter minima from SAPT intrinsically improve test-time sample selection.

**Ablative analysis on key parameter** $\rho'$**.** Theoretical analysis (Section 4) establishes $\rho$ and $\rho'$ as key generalization controllers, playing significant roles during the training and testing stages, respectively. Since $\rho$'s role has been well explored in previous research (Foret et al., 2020), we focus on $\rho'$ for test-time adaptation: as mentioned earlier, it governs distribution distinguishability, with proper values enhancing prediction reliability through sensitive discrimination. Empirical validation on ImageNet-R (Figure 3) shows non-monotonic accuracy dependence on $\rho'$—initially rising then falling. It is because extreme values ($\rho' \rightarrow 0$ or $\rho' \gg 0$) may yield uninformative sharpness measures and degrade performance. Crucially, $\rho' = 0$ degenerates to entropy maximization, and its comparison with non-zero cases also demonstrates sharpness's necessity. Notably, all experiments fix $\rho' = 0.5$ without test-data tuning, and this analysis solely aims to demonstrate the control effect of $\rho'$ on generalization. Additionally, sample retention follows a similar trend: accuracy peaks then declines with increased retention. This reflects the probabilistic correlation: lower sharpness typically means greater proximity to the training distribution, meaning performance degrades when retaining excessively high-sharpness samples.

## 5.4 Visualization of Loss Landscapes

To intuitively validate FGA's effectiveness, we visualize the test data's loss surface using a 2D technique (Li et al., 2018) in Figure 4, revealing how sample selection enhances prediction reliability. The visualization demonstrates critical relationships: when parameters reside in flat minima (Figure 4, top), augmented samples maintain semantic integrity and enable reliable predictions. Conversely, parameters outside flat minima (Figure 4, bottom) yield distorted semantic representations that degrade generalization. This contrast clearly illustrates that FGA can help filter out unreliable test samples to mitigate their adverse effects, thereby strengthening the generalization ability of VLMs.

## 6 Conclusion

This paper demonstrates that flatness operates not just as a beneficial training characteristic but as a key geometric clue for test-time adaptation. This understanding motivates the proposal of a novel framework, Flatness-Guided Adaptation (FGA), which utilizes the principle of loss landscape flatness as a unified guide to improve both training and test generalization against distribution shifts. Different from previous TTA methods that often fine-tune prompts per sample, it directs adaptation by leveraging the geometric relationship between training minima and test-time loss landscapes. Specifically, it first identifies flat minima during prompt tuning and then ensures the alignment across training and test landscapes via a selective mechanism. Comprehensive experiments and theoretical analysis confirm FGA's effectiveness and superior performance. We anticipate this work will advance the understanding of loss landscapes and inspire future TTA technologies.

**Future work.** Our proposed FGA is grounded in a general analysis of loss landscape geometry, a foundational concept that is broadly applicable across diverse model architectures and learning paradigms. This foundation makes FGA a flexible component that could be integrated into modern visual-language models, advanced prompt tuning methods, or new types of test-time adaptation objectives to potentially enhance their performance. In this work, we intentionally followed the experimental setup introduced in the TPT paper. This choice allows a controlled and fair comparison with prior TTA methods, helping us to clearly demonstrate the contribution of FGA itself. The extension of FGA to other experimental configurations would require extensive engineering efforts to conduct large-scale experiments across diverse settings, and thus remains our future work.

## Acknowledgements

This work was financially supported by the National Natural Science Foundation of China (NSFC) under Grant No.U20B2070. The authors gratefully acknowledge the NSFC for its critical support of this research.

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
