# SUPPLEMENTARY MATERIAL FOR "FLATNESS GUIDED TEST-TIME ADAPTATION FOR VISION-LANGUAGE MODELS"

This supplementary document provides rigorous technical foundations and extended experimental validation for the main paper "Flatness Guided Test-Time Adaptation for Vision-Language Models". Section A and B establishes the theoretical bedrock through complete proofs of Theorem 1 and Theorem 4 (stated in the main text), formally deriving the generalization error bounds via the divergence discrepancy and Rademacher complexity analysis while elucidating the roles of hyperparameters $\rho$ and $\rho'$ in loss landscape smoothing and test distribution discrimination. Section C presents comprehensive experimental results on both domain generalization and cross-dataset benchmarks with CLIP-ResNet50 architecture (see Tables 1-2). Then, Section D delivers full reproducibility resources, specifying critical hyperparameters, code architecture, and hardware configurations to enable exact replication. Finally, the document ends with a note on LLM utilization (in Section E) for language refinement.

## A  PROOF OF THEOREM 1

We begin by defining the divergence between two distributions, which is crucial for bounding the difference in their corresponding loss functions.

**Definition A.1.** *For two probability distributions $\mathcal{S}$ and $\mathcal{T}$ defined on a common measurable space, the divergence $d(\mathcal{S}, \mathcal{T})$ is defined as twice the supremum of the absolute difference between their probabilities over all measurable subsets A, i.e.,*

$$d(\mathcal{S}, \mathcal{T}) := 2 \sup_A |\mathbb{P}_\mathcal{S}(A) - \mathbb{P}_\mathcal{T}(A)|, \tag{1}$$

*where $\mathbb{P}_\mathcal{S}(A)$ and $\mathbb{P}_\mathcal{T}(A)$ represent the probabilities of event $A$ under distributions $\mathcal{S}$ and $\mathcal{T}$, and $\sup_A$ denotes the supremum taken over all such subsets $A$.*

With this definition in hand, we now move to a lemma that connects the distance between distributions to the difference in the losses, setting the stage for bounding the generalization error.

**Lemma A.2** ((Cha et al., 2021)). *Consider an bounded loss function $\ell : \mathbb{R}^K \times \mathbb{R} \to [0, M]$. Given two distributions, $\mathcal{S}$ and $\mathcal{T}$, the difference between the loss with $\mathcal{S}$ and the loss with $\mathcal{T}$ is bounded by the distance between $\mathcal{S}$ and $\mathcal{T}$:*

$$|\mathbb{E}_\mathcal{T}\ell(\boldsymbol{f}_{\boldsymbol{\theta}}(\boldsymbol{x}_\tau), y_\tau) - \mathbb{E}_\mathcal{S}\ell(\boldsymbol{f}_{\boldsymbol{\theta}}(\boldsymbol{x}_s), y_s)| \leq \frac{M}{2} d(\mathcal{S}; \mathcal{T}). \tag{2}$$

*Proof.* Firstly, using the identity that for any non-negative random variable $X$, $\mathbb{E}[X] = \int_0^\infty \mathbb{P}(X > t)dt$, we rewrite the difference as an integral over the tail probabilities of the loss:

$$|\mathbb{E}_\mathcal{T}\ell(\boldsymbol{f}_{\boldsymbol{\theta}}(\boldsymbol{x}_\tau), y_\tau) - \mathbb{E}_\mathcal{S}\ell(\boldsymbol{f}_{\boldsymbol{\theta}}(\boldsymbol{x}_s), y_s)|$$

$$= \left| \int_0^\infty \mathbb{P}_\mathcal{T}(\ell(\boldsymbol{f}_{\boldsymbol{\theta}}(\boldsymbol{x}_\tau), y_\tau) > t)\, dt - \int_0^\infty \mathbb{P}_\mathcal{S}(\ell(\boldsymbol{f}_{\boldsymbol{\theta}}(\boldsymbol{x}_s), y_s) > t)\, dt \right|$$

$$\leq \int_0^\infty |\mathbb{P}_\mathcal{T}(\ell(\boldsymbol{f}_{\boldsymbol{\theta}}(\boldsymbol{x}_\tau), y_\tau) > t) - \mathbb{P}_\mathcal{S}(\ell(\boldsymbol{f}_{\boldsymbol{\theta}}(\boldsymbol{x}_s), y_s) > t)|\, dt. \tag{3}$$

Since the loss $\ell$ is bounded in $[0, M]$, the tail probability is zero for all $t > M$. Thus, the above integral simplifies to:

$$I = \int_0^M |\mathbb{P}_\mathcal{T}(\ell(\boldsymbol{f}_{\boldsymbol{\theta}}(\boldsymbol{x}_\tau), y_\tau) > t) - \mathbb{P}_\mathcal{S}(\ell(\boldsymbol{f}_{\boldsymbol{\theta}}(\boldsymbol{x}_s), y_s) > t)|\, dt. \tag{4}$$

This integral is bounded above by taking the supremum over all $t \in [0, M]$ and all hypotheses $\boldsymbol{f_\theta}$:

$$I \leq M \cdot \sup_{t \in [0,M]} \sup_{\boldsymbol{f_\theta}} |\mathbb{P}_{\mathcal{T}} \left( \ell(\boldsymbol{f_\theta}(\boldsymbol{x}_\tau), y_\tau) > t \right) - \mathbb{P}_{\mathcal{S}} \left( \ell(\boldsymbol{f_\theta}(\boldsymbol{x}_s), y_s) > t \right)|. \tag{5}$$

Now, for each fixed $t$ and $\boldsymbol{f_\theta}$, the set $\{\ell(\boldsymbol{f_\theta}(\boldsymbol{x}_s), y_s) > t\}$ is a measurable event; hence the double supremum over such sets is dominated by the supremum over all measurable events $A$. Consequently, we obtain:

$$|\mathbb{E}_{\mathcal{T}} \ell\left(\boldsymbol{f_\theta}(\boldsymbol{x}_\tau), y_\tau\right) - \mathbb{E}_{\mathcal{S}} \ell\left(\boldsymbol{f_\theta}(\boldsymbol{x}_s), y_s\right)| \leq M \cdot \sup_A |\mathbb{P}_{\mathcal{T}}(A) - \mathbb{P}_{\mathcal{S}}(A)|. \tag{6}$$

By referring back to Definition 1, we recognize that this supremum equals half the divergence $d(\mathcal{S}, \mathcal{T})$, which completes the proof.

$\square$

Next, we incorporate the Rademacher complexity bound for the generalization error. This bound will provide a crucial link between the expected loss and the sample complexity.

**Lemma A.3** (Theorem 6.31 (Zhang, 2023)). *Consider a real-valued function class $\mathcal{F} = \{f_{\boldsymbol{\theta}}(\cdot)\}$ and a bounded loss function $\ell : \mathbb{R} \times \mathbb{R} \to [0, M]$. Assume that the loss function $\ell(f, y)$ is $\mu$-Lipschitz with respect to $f$ :*

$$|\ell(f, y) - \ell(f', y)| \leq \mu |f - f'|. \tag{7}$$

*Let $\mathcal{S}_n = \{\boldsymbol{x}_i, y_i\}_{i=1}^n$ be $n$ i.i.d. samples from distribution $\mathcal{S}$. With probability at least $1 - \delta$:*

$$\mathbb{E}_{\mathcal{S}} \ell(f_{\boldsymbol{\theta}}(\boldsymbol{x}), y) \leq \frac{1}{n} \sum_{i=1}^n \ell\left(f_{\boldsymbol{\theta}}(\boldsymbol{x}_i), y_i\right) + 2\mu R_n(\mathcal{F}, \mathcal{S}) + M \sqrt{\frac{\log(1/\delta)}{2n}}. \tag{8}$$

*Here, $R_n(\mathcal{F}, \mathcal{S})$ represents the expected Rademacher complexity:*

$$R_n(\mathcal{F}, \mathcal{S}) = \mathbb{E}_{(\boldsymbol{x}_i, y_i) \sim \mathcal{S}} \mathbb{E}_\sigma \sup_{f \in \mathcal{F}} \frac{1}{n} \sum_{i=1}^n \sigma_i f_{\boldsymbol{\theta}}(\boldsymbol{x}_i), \tag{9}$$

*where $\sigma_1, \ldots, \sigma_n$ are independent uniform $\{\pm 1\}$-valued Bernoulli random variables.*

The lemma above, which addresses real-valued function classes, can be readily extended to the setting of vector-valued function classes (Theorem A.5). Before presenting its formal statement and proof, we first introduce a necessary supporting lemma (Lemma A.4).

**Lemma A.4** (Corollary 4 (Maurer, 2016)). *Let $\mathcal{X}$ be any set, $(\boldsymbol{x}_1, \ldots, \boldsymbol{x}_n) \in \mathcal{X}^n$, let $\mathcal{F}_v$ be a class of functions $\boldsymbol{f} : \mathcal{X} \to \mathbb{R}^K$ and let $h_i : \mathbb{R}^K \to \mathbb{R}$ have Lipschitz norm $\mu$. Then*

$$\mathbb{E} \sup_{\boldsymbol{f} \in F_v} \sum_{i=1}^n \sigma_i h_i\left(\boldsymbol{f}(\boldsymbol{x}_i)\right) \leq \sqrt{2}\mu \, \mathbb{E} \sup_{\boldsymbol{f} \in F_v} \sum_{i=1}^n \sum_k \sigma_{ik} f_k(x_i), \tag{10}$$

*where $\{\sigma_i\}$ and $\{\sigma_{ik}\}$ are independent Bernoulli sequences, and $f_k(\boldsymbol{x}_i)$ denotes the $k$-th coordinate of $\boldsymbol{f}(\boldsymbol{x}_i) \in \mathbb{R}^K$.*

**Theorem A.5.** *Consider a vector-valued function class $\mathcal{F}_v = \{\boldsymbol{f_\theta}(\cdot) : \mathcal{X} \to \mathbb{R}^K\}$ and a bounded loss function $\ell : \mathbb{R}^K \times \mathcal{Y} \to [0, M]$. Assume that the loss function $\ell(\boldsymbol{f}, y)$ is $\mu$-Lipschitz with respect to $\boldsymbol{f}$. Let $\mathcal{S}_n = \{\boldsymbol{x}_i, y_i\}_{i=1}^n$ be $n$ i.i.d. samples from distribution $\mathcal{S}$. With probability at least $1 - \delta$:*

$$\mathbb{E}_{\mathcal{S}} \ell(\boldsymbol{f_\theta}(\boldsymbol{x}), y) \leq \frac{1}{n} \sum_{i=1}^n \ell\left(f_{\boldsymbol{\theta}}(\boldsymbol{x}_i), y_i\right) + 2\sqrt{2}\mu R_n(\mathcal{F}_v, \mathcal{S}) + M \sqrt{\frac{\log(1/\delta)}{2n}}. \tag{11}$$

*Here, $R_n(\mathcal{F}_v, \mathcal{S})$ represents the expected Rademacher complexity:*

$$R_n(\mathcal{F}_v, \mathcal{S}) = \mathbb{E}_{(\boldsymbol{x}_i, y_i) \sim \mathcal{S}} \mathbb{E}_\sigma \sup_{\boldsymbol{f} \in \mathcal{F}_v} \frac{1}{n} \sum_{i=1}^n \boldsymbol{\sigma}_i^\top \boldsymbol{f_\theta}(\boldsymbol{x}_i), \tag{12}$$

*where $\boldsymbol{\sigma}_1, \ldots, \boldsymbol{\sigma}_n$ are independent uniform $\{\pm 1\}$-valued Bernoulli random vectors.*

*Proof.* Considering the real-valued function class $\mathcal{G} = \{g = \ell \circ \boldsymbol{f_\theta} : \mathcal{X} \to \mathbb{R}\}$ and the identity map $h : \mathbb{R} \to \mathbb{R}$, which has Lipschitz norm 1, we apply Lemma A.3 to obtain that, with probability at least $1 - \delta$,

$$\mathbb{E}_{\mathcal{S}}\big[\ell(f_{\boldsymbol{\theta}}(\boldsymbol{x}), y)\big] \leq \frac{1}{n}\sum_{i=1}^{n}\ell\big(f_{\boldsymbol{\theta}}(\boldsymbol{x}_i), y_i\big) + 2R_n(\mathcal{G}, \mathcal{S}) + M\sqrt{\frac{\log(1/\delta)}{2n}}, \tag{13}$$

where $R_n(\mathcal{G}, \mathcal{S})$ denotes the expected Rademacher complexity of $\mathcal{G}$ over $\mathcal{S}$:

$$R_n(\mathcal{G}, \mathcal{S}) = \mathbb{E}\sup_{\ell \circ \boldsymbol{f} \in \mathcal{G}}\frac{1}{n}\sum_{i=1}^{n}\sigma_i \ell(\boldsymbol{f_\theta}(\boldsymbol{x}_i), y_i) \leq \sqrt{2}\,\mu\,\mathbb{E}\sup_{\boldsymbol{f_\theta} \in \mathcal{F}}\frac{1}{n}\sum_{i=1}^{n}\boldsymbol{\sigma}_i^\top \boldsymbol{f_\theta}(\boldsymbol{x}_i). \tag{14}$$

The last inequality follows from Lemma A.4. $\qquad\square$

Now, by combining Lemma A.2 and Lemma A.5, we conclude that with probability at least $1 - \delta$,

$$\mathbb{E}_{\mathcal{T}}\big[\ell^\rho(\boldsymbol{f_\theta}(\boldsymbol{x}_\mathcal{T}), y_\mathcal{T})\big] \leq \frac{M}{2}\,d\,(\mathcal{S};\mathcal{T}) + \hat{\ell}^\rho_{\mathcal{S}_n}(\boldsymbol{f_\theta}) + 2\sqrt{2}\,\mu\,R_n(\mathcal{F}_v, \mathcal{S}) + M\sqrt{\frac{\log(1/\delta)}{2n}}, \tag{15}$$

where $\hat{\ell}^\rho_{\mathcal{S}_n}(\boldsymbol{f_\theta}) := \frac{1}{n}\sum_{i=1}^{n}\ell^\rho\big(\boldsymbol{f_\theta}(\boldsymbol{x}_i), y_i\big)$ denotes the empirical loss over the training set $\mathcal{S}_n$. This completes the proof of Theorem 1.

*Remarks.* The generalization error bound above does not explicitly include $\rho$, which could be a significant parameter for mitigating generalization error. However, it is important to note that $\rho$ can enhance the smoothness of $\ell^\rho$, thereby potentially reducing the Lipschitz constant $\mu$, which is a contributing factor to the expected Rademacher complexity $R_n(\mathcal{F}_v, \mathcal{S})$.

## B  PROOF OF THEOREM 4

We begin by proving a simplified version of Theorem 4, stated as follows.

**Theorem B.1.** *Consider a vector-valued function class $\mathcal{F}_v = \{\boldsymbol{f_\theta}(\cdot) : \mathcal{X} \to \mathbb{R}^K\}$, where the parameters $\boldsymbol{\theta}$ lie in a set $\Theta$ such that the loss function $\ell$ is bounded within $[0, M]$. Given a training distribution $\mathcal{S}$ and two $\gamma$-separable test distributions $\mathcal{T}_1$ and $\mathcal{T}_2$, assume $d(\mathcal{S}, \mathcal{T}_1) < d(\mathcal{S}, \mathcal{T}_2)$. Define the quantile function $q_i(\delta)$ for the entropy loss of $\boldsymbol{f_\theta}$ on $\mathcal{T}_i$ such that $\mathbb{P}\left(H^\rho < \mathbb{E}[H^\rho] + q_i(\delta)\right) = 1 - \delta$, and let $Q(\delta) = \sup\{Q_1(\delta), Q_2(\delta)\}$ be the supremum quantile. Then, with probability at least $1 - \delta$:*

$$\ell^\rho(\boldsymbol{f_\theta}(\boldsymbol{x}_{\mathcal{T}_i}), y_{\mathcal{T}_i}) \leq \frac{M}{2}d(\mathcal{S}, \mathcal{T}_i) + \hat{\ell}^\rho_{\mathcal{S}}(\boldsymbol{f_\theta}) + 2\sqrt{2}\,\mu\,R_n(\mathcal{F}_v, \mathcal{S}) + M\sqrt{\frac{\log(2/\delta)}{2n}} + Q(\delta/2). \tag{16}$$

*If this bound is $\beta_i$-tight for $\mathcal{T}_1$ and $\mathcal{T}_2$ with $\beta_i < \gamma$, then there exists a threshold $\xi$ such that:*

$$\mathbb{P}\left(\ell^\rho(\boldsymbol{f_\theta}(\boldsymbol{x}_{\mathcal{T}_1}), y_{\mathcal{T}_1}) < \xi\right) > \mathbb{P}\left(\ell^\rho(\boldsymbol{f_\theta}(\boldsymbol{x}_{\mathcal{T}_2}), y_{\mathcal{T}_2}) < \xi\right). \tag{17}$$

*Proof.* By Theorem 1, with probability at least $1 - \delta/2$:

$$\mathbb{E}_{\mathcal{T}_i}[\ell^\rho(\boldsymbol{f_\theta}(\boldsymbol{x}_{\mathcal{T}_i}), y_{\mathcal{T}_i})] \leq \hat{\ell}^\rho_{\mathcal{S}_n}(\boldsymbol{f_\theta}) + \frac{M}{2}d\,(\mathcal{S};\mathcal{T}_i) + 2\sqrt{2}\,\mu\,R_n(\mathcal{F}_v, \mathcal{S}) + M\sqrt{\frac{\log(2/\delta)}{2n}}. \tag{18}$$

Furthermore, by the definition of the quantile function, with probability at least $1 - \delta/2$:

$$\ell^\rho(\boldsymbol{f_\theta}(\boldsymbol{x}_{\mathcal{T}_i}), y_{\mathcal{T}_i}) \leq \mathbb{E}_{\mathcal{T}_i}[\ell^\rho(\boldsymbol{f_\theta}(\boldsymbol{x}_{\mathcal{T}_i}), y_{\mathcal{T}_i})] + Q_i(\delta/2). \tag{19}$$

Combining these two results yields, with probability at least $1 - \delta$:

$$\ell^\rho(\boldsymbol{f_\theta}(\boldsymbol{x}_{\mathcal{T}_i}), y_{\mathcal{T}_i}) \leq \frac{M}{2}d(\mathcal{S}, \mathcal{T}_i) + \hat{\ell}^\rho_{\mathcal{S}}(\boldsymbol{f_\theta}) + 2\sqrt{2}\,\mu\,R_n(\mathcal{F}_v, \mathcal{S}) + M\sqrt{\frac{\log(2/\delta)}{2n}} + Q_i(\delta/2). \tag{20}$$

Since $Q(\delta) = \sup\{Q_1(\delta), Q_2(\delta)\}$ is the supremum quantile, it follows that

$$\ell^\rho(\boldsymbol{f_\theta}(\boldsymbol{x}_{\mathcal{T}_i}), y_{\mathcal{T}_i}) \leq \frac{M}{2}d(\mathcal{S}, \mathcal{T}_i) + \hat{\ell}^\rho_{\mathcal{S}}(\boldsymbol{f_\theta}) + 2\sqrt{2}\,\mu\,R_n(\mathcal{F}_v, \mathcal{S}) + M\sqrt{\frac{\log(2/\delta)}{2n}} + Q(\delta/2). \tag{21}$$

Then, for the subsequent analysis, we define two quantities, $\xi_1$ and $\xi_2$, as follows:

$$\xi_{1,2} := \frac{M}{2} d\left(\mathcal{S}; \mathcal{T}_{1,2}\right) + Q(\delta/2) + \hat{\ell}_{\mathcal{S}}^{\rho}(\boldsymbol{f_\theta}) + 2\sqrt{2}\,\mu\,R_n(\mathcal{F}_v, \mathcal{S}) + M\sqrt{\frac{\log(2/\delta)}{2n}}. \quad (22)$$

Next, we consider the relationship between the divergences $d(\mathcal{S}; \mathcal{T}_1)$ and $d(\mathcal{S}; \mathcal{T}_2)$. If $d(\mathcal{S}; \mathcal{T}_1) < d(\mathcal{S}; \mathcal{T}_2)$, then we immediately conclude that $\xi_1 < \xi_2$. From Theorem 1, we can further derive the following two inequalities:

$$\mathbb{P}\left(\ell^{\rho}(\boldsymbol{f_\theta}(\boldsymbol{x}_{\mathcal{T}_1}), y_{\mathcal{T}_1}) > \xi_1\right) < \delta, \quad (23)$$

and

$$\mathbb{P}\left(\ell^{\rho}(\boldsymbol{f_\theta}(\boldsymbol{x}_{\mathcal{T}_2}), y_{\mathcal{T}_2}) > \xi_2\right) < \delta. \quad (24)$$

To proceed, let us introduce $\xi^{\star}$, the oracle upper bound of $\ell^{\rho}(\boldsymbol{f_\theta}(\boldsymbol{x}_{\mathcal{T}_2}), y_{\mathcal{T}_2})$, which satisfies:

$$\mathbb{P}\left(\ell^{\rho}(\boldsymbol{f_\theta}(\boldsymbol{x}_{\mathcal{T}_2}), y_{\mathcal{T}_2}) > \xi^{\star}\right) = \delta. \quad (25)$$

Next, we define the difference $\beta$ as the absolute difference between $\xi_2$ and the oracle bound $\xi^{\star}$, i.e., $\beta := |\xi_2 - \xi^{\star}|$. Given the separability of $\mathcal{T}_1$ and $\mathcal{T}_2$, we can infer that $|\xi_2 - \xi_1| > \beta$. This implies that: $\xi_1 < \xi_2 - \beta \leq \xi^{\star}$. Consequently, we can conclude that:

$$\mathbb{P}\left(\ell^{\rho}(\boldsymbol{f_\theta}(\boldsymbol{x}_{\mathcal{T}_2}), y_{\mathcal{T}_2}) > \xi_1\right) > \mathbb{P}\left(\ell^{\rho}(\boldsymbol{f_\theta}(\boldsymbol{x}_{\mathcal{T}_2}), y_{\mathcal{T}_2}) > \xi^{\star}\right). \quad (26)$$

Now, let us set $\xi = \xi_1$, which leads us to the following inequalities:

$$\mathbb{P}\left(\ell^{\rho}(\boldsymbol{f_\theta}(\boldsymbol{x}_{\mathcal{T}_1}), y_{\mathcal{T}_1}) > \xi\right) < \delta < \mathbb{P}\left(\ell^{\rho}(\boldsymbol{f_\theta}(\boldsymbol{x}_{\mathcal{T}_2}), y_{\mathcal{T}_2}) > \xi_1\right), \quad (27)$$

which completes the proof of Theorem B.1. $\qquad\square$

Please note that in the absence of true labels during test-time adaptation, we must employ entropy as a surrogate loss for cross entropy, which is equivalent to using cross entropy with conjugate pseudo-labels and is considered the best alternative to cross entropy(Goyal et al., 2022). However, using a surrogate loss introduces an additional error term, which relates to the following theorem.

**Theorem B.2** (Upper bound on the difference between entropy and cross entropy). *Let $\boldsymbol{p} = (p_1, \ldots, p_K)$ and $\boldsymbol{q} = (q_1, \ldots, q_K)$ be two probability distributions over a finite set $\{1, \ldots, K\}$. Suppose there exists a constant $\eta > 0$ such that $p_i, q_i \geq \eta$ for all $i$. Then the entropy $H(\boldsymbol{q})$ and the cross entropy $H(\boldsymbol{p}, \boldsymbol{q})$ satisfy the following inequality:*

$$|H(\boldsymbol{p}, \boldsymbol{q}) - H(\boldsymbol{q})| \leq \log\frac{1}{\eta} \cdot \|\boldsymbol{p} - \boldsymbol{q}\|_1 + \frac{1}{\eta}\|\boldsymbol{p} - \boldsymbol{q}\|_1^2, \quad (28)$$

*where $\|\boldsymbol{p} - \boldsymbol{q}\|_1 = \sum_{i=1}^{K} |p_i - q_i|$ is the $L^1$ distance between the distributions.*

*Proof.* We begin by recalling the relevant definitions. The entropy of $q$ is $H(\boldsymbol{q}) = -\sum_{i=1}^{K} q_i \log q_i$, and the cross entropy between $p$ and $q$ is $H(\boldsymbol{p}, \boldsymbol{q}) = -\sum_{i=1}^{K} p_i \log q_i$. The Kullback–Leibler divergence is defined as $D_{\mathrm{KL}}(\boldsymbol{p}\|\boldsymbol{q}) = \sum_{i=1}^{K} p_i \log \frac{p_i}{q_i}$. A well-known identity relates these quantities:

$$H(\boldsymbol{p}, \boldsymbol{q}) = H(\boldsymbol{p}) + D_{\mathrm{KL}}(\boldsymbol{p}\|\boldsymbol{q}). \quad (29)$$

We can express the difference of interest as:

$$H(\boldsymbol{p}, \boldsymbol{q}) - H(\boldsymbol{q}) = [H(\boldsymbol{p}, \boldsymbol{q}) - H(\boldsymbol{p})] + [H(\boldsymbol{p}) - H(\boldsymbol{q})] = D_{\mathrm{KL}}(\boldsymbol{p}\|\boldsymbol{q}) + [H(\boldsymbol{p}) - H(\boldsymbol{q})]. \quad (30)$$

Taking absolute values and applying the triangle inequality yields:

$$|H(\boldsymbol{p}, \boldsymbol{q}) - H(\boldsymbol{q})| \leq D_{\mathrm{KL}}(\boldsymbol{p}\|\boldsymbol{q}) + |H(\boldsymbol{p}) - H(\boldsymbol{q})|. \quad (31)$$

The rest of the proof consists of bounding the two terms on the right-hand side.

First, we bound the KL divergence. We establish the upper bound for the KL divergence using a second-order Taylor expansion along the linear path connecting the two distributions. Define the parametric family of distributions $\boldsymbol{r}(t) = (1-t)\boldsymbol{q} + t\boldsymbol{p}$ for $t \in [0, 1]$, which interpolates between $\boldsymbol{r}(0) = \boldsymbol{q}$ and $\boldsymbol{r}(1) = \boldsymbol{p}$. Consider the function $f(t) = D_{\mathrm{KL}}(\boldsymbol{r}(t)\|\boldsymbol{q})$, which measures the KL

divergence from the interpolated distribution to the target distribution $\boldsymbol{q}$. Our goal is to compute $f(1) = D_{\mathrm{KL}}(\boldsymbol{p}\|\boldsymbol{q})$ using Taylor's theorem. We begin by computing the derivatives of $f(t)$:

$$f'(t) = \sum_i (p_i - q_i) \log \frac{r_i(t)}{q_i}, \quad f''(t) = \sum_i \frac{(p_i - q_i)^2}{r_i(t)}. \tag{32}$$

We now apply Taylor's theorem with Lagrange remainder at $t = 0$. Since $f(0) = D_{\mathrm{KL}}(\boldsymbol{q}\|\boldsymbol{q}) = 0$ and $f'(0) = \sum_i (p_i - q_i) \log \frac{q_i}{q_i} = 0$, the expansion yields $f(t) = \frac{1}{2}f''(s)t^2$ for some $s \in [0, t]$. Setting $t = 1$ gives the key identity $D_{\mathrm{KL}}(\boldsymbol{p}\|\boldsymbol{q}) = \frac{1}{2}\sum_i \frac{(p_i - q_i)^2}{r_i(s)}$, where $r_i(s) = (1 - s)q_i + sp_i$. The critical step is to bound the denominator $r_i(s)$ away from zero. Since both $p_i \geq \eta$ and $q_i \geq \eta$ by assumption, and since $r_i(s)$ is a convex combination of $p_i$ and $q_i$, we have $r_i(s) \geq \eta$ for all $i$ and all $s \in [0, 1]$. This allows us to bound each term in the sum by $\frac{(p_i - q_i)^2}{r_i(s)} \leq \frac{(p_i - q_i)^2}{\eta}$. Summing over all components gives $\sum_i \frac{(p_i - q_i)^2}{r_i(s)} \leq \frac{1}{\eta}\sum_i (p_i - q_i)^2$. Then, we relate the sum of squares to the $L^1$ distance. For each $i$, we have $(p_i - q_i)^2 \leq |p_i - q_i| \cdot \|\boldsymbol{p} - \boldsymbol{q}\|_1$ because $|p_i - q_i| \leq \|\boldsymbol{p} - \boldsymbol{q}\|_1$. Summing this inequality over $i$ gives $\sum_i (p_i - q_i)^2 \leq \|\boldsymbol{p} - \boldsymbol{q}\|_1^2$. Combining this with the previous bound, we obtain:

$$D_{\mathrm{KL}}(\boldsymbol{p}\|\boldsymbol{q}) \leq \frac{1}{2\eta}\|\boldsymbol{p} - \boldsymbol{q}\|_1^2. \tag{33}$$

Next, we bound the difference $|H(\boldsymbol{p}) - H(\boldsymbol{q})|$. Consider the entropy function $H(\boldsymbol{r}) = -\sum_{i=1}^{K} r_i \log r_i$, defined on the probability simplex. Its gradient has components $\frac{\partial H}{\partial r_i} = -\log r_i - 1$, and its Hessian is a diagonal matrix with entries $-\frac{1}{r_i}$ on the diagonal. We perform a second-order Taylor expansion of $H(\boldsymbol{q})$ around $\boldsymbol{p}$:

$$H(\boldsymbol{q}) = H(\boldsymbol{p}) + \langle \nabla H(\boldsymbol{p}), \boldsymbol{q} - \boldsymbol{p} \rangle + \frac{1}{2}(\boldsymbol{q} - \boldsymbol{p})^\top \nabla^2 H(\boldsymbol{\xi})(\boldsymbol{q} - \boldsymbol{p}), \tag{34}$$

where $\boldsymbol{\xi}$ lies on the segment between $\boldsymbol{p}$ and $\boldsymbol{q}$. Since $p_i, q_i \geq \eta$ and the simplex is convex, we also have $\xi_i \geq \eta$. We now bound each term. The linear term satisfies:

$$|\langle \nabla H(\boldsymbol{p}), \boldsymbol{q} - \boldsymbol{p} \rangle| = |\langle \nabla \tilde{H}(\boldsymbol{p}), \boldsymbol{q} - \boldsymbol{p} \rangle| \leq \|\nabla \tilde{H}(\boldsymbol{p})\|_\infty \cdot \|\boldsymbol{q} - \boldsymbol{p}\|_1. \tag{35}$$

Here, since $\sum_i (q_i - p_i) = 0$, we use the notation $\nabla \tilde{H}(\boldsymbol{p})$ to denote $\nabla H(\boldsymbol{p}) + 1$. Because $\|\nabla \tilde{H}(\boldsymbol{p})\|_\infty = \max_{1 \leq j \leq K} |\log p_j| = \max_{1 \leq j \leq K} |\log \frac{1}{p_j}| \leq \log \frac{1}{\eta}$, we have

$$|\langle \nabla H(\boldsymbol{p}), \boldsymbol{q} - \boldsymbol{p} \rangle| \leq \log \frac{1}{\eta} \cdot \|\boldsymbol{p} - \boldsymbol{q}\|_1. \tag{36}$$

For the quadratic term, the Hessian satisfies $|(\boldsymbol{q} - \boldsymbol{p})^\top \nabla^2 H(\boldsymbol{\xi})(\boldsymbol{q} - \boldsymbol{p})| \leq \frac{1}{\eta}\|\boldsymbol{q} - \boldsymbol{p}\|_2^2$. Since $\|\boldsymbol{q} - \boldsymbol{p}\|_2^2 \leq \|\boldsymbol{q} - \boldsymbol{p}\|_1^2$, we obtain:

$$\left|\frac{1}{2}(\boldsymbol{q} - \boldsymbol{p})^\top \nabla^2 H(\boldsymbol{\xi})(\boldsymbol{q} - \boldsymbol{p})\right| \leq \frac{1}{2\eta}\|\boldsymbol{p} - \boldsymbol{q}\|_1^2. \tag{37}$$

Combining (36) and (37), we conclude:

$$|H(\boldsymbol{p}) - H(\boldsymbol{q})| \leq \log \frac{1}{\eta} \cdot \|\boldsymbol{p} - \boldsymbol{q}\|_1 + \frac{1}{2\eta}\|\boldsymbol{p} - \boldsymbol{q}\|_1^2. \tag{38}$$

Substituting the bounds (33) and (38) into inequality (31) gives:

$$|H(\boldsymbol{p}, \boldsymbol{q}) - H(\boldsymbol{q})| \leq \log \frac{1}{\eta} \cdot \|\boldsymbol{p} - \boldsymbol{q}\|_1 + \frac{1}{\eta}\|\boldsymbol{p} - \boldsymbol{q}\|_1^2, \tag{39}$$

which completes the proof. $\square$

A direct application of Theorem B.1 and Theorem B.2 yields the following corollary, which is formally stated as Theorem 4.

**Corollary B.3.** *Consider a vector-valued function class $\mathcal{F}_v = \{\boldsymbol{f_\theta}(\cdot) : \mathcal{X} \to \mathbb{R}^K\}$, where the parameters $\boldsymbol{\theta}$ lie in a set $\Theta$ such that the loss function is bounded within $[0, M]$. Let $\boldsymbol{f} = (f_1, \ldots, f_K)$ and $\mathbf{y} = (\mathrm{y}_1, \ldots, \mathrm{y}_K)$ be probability distributions over a finite set $\{1, \ldots, K\}$, with $f_i, \mathrm{y}_i \geq \eta > 0$ for all $i$. Denote by $H(\boldsymbol{f})$ the entropy and by $H(\mathbf{y}, \boldsymbol{f})$ the cross entropy. Assume the worst-case cross entropy $H^\rho(\mathbf{y}, \boldsymbol{f})$ is $\mu$-Lipschitz continuous with respect to $\boldsymbol{f}$ over $\Theta$. Given a training distribution $\mathcal{S}$ and two $\gamma$-separable test distributions $\mathcal{T}_1$ and $\mathcal{T}_2$, assume $d(\mathcal{S}, \mathcal{T}_1) < d(\mathcal{S}, \mathcal{T}_2)$. Define the quantile function $Q_i(\delta)$ for the entropy loss of $\boldsymbol{f_\theta}$ on $\mathcal{T}_i$ such that $\mathbb{P}(H^\rho < \mathbb{E}[H^\rho] + Q_i(\delta)) = 1 - \delta$, and let $Q(\delta) = \sup\{Q_1(\delta), Q_2(\delta)\}$ be the supremum quantile. Then, with probability at least $1 - \delta$:*

$$H^\rho(\boldsymbol{f_\theta}(\boldsymbol{x}_{\mathcal{T}_i})) \leq \frac{M}{2} d(\mathcal{S}; \mathcal{T}_i) + Q(\delta/2) + \hat{H}^\rho_{\mathcal{S}_n} + 2\mu R_n(\mathcal{F}_v, \mathcal{S}) + M\sqrt{\frac{\log(2/\delta)}{2n}}$$
$$+ \log\frac{1}{\eta} \cdot \mathbb{E}_\mathcal{S}\|\mathbf{y}_\mathcal{S} - \boldsymbol{f}_{\widetilde{\boldsymbol{\theta}}}(\boldsymbol{x}_\mathcal{S})\|_1 + \frac{1}{\eta}\mathbb{E}_\mathcal{S}\|\mathbf{y}_\mathcal{S} - \boldsymbol{f}_{\widetilde{\boldsymbol{\theta}}}(\boldsymbol{x}_\mathcal{S})\|_1^2. \tag{40}$$

*Here, the notation $\widetilde{\boldsymbol{\theta}}$ is defined as $\widetilde{\boldsymbol{\theta}} := \boldsymbol{\theta} + \arg\max_{\|\boldsymbol{\epsilon}\| \leq \rho} \max\{H(\mathbf{y}, \boldsymbol{f}_{\boldsymbol{\theta}+\boldsymbol{\epsilon}}(\boldsymbol{x})), H(\boldsymbol{f}_{\boldsymbol{\theta}+\boldsymbol{\epsilon}}(\boldsymbol{x}))\}$. Furthermore, if this bound is $\beta_i$-tight for $\mathcal{T}_1$ and $\mathcal{T}_2$ with $\beta_i < \gamma$, then there exists a threshold $\xi$ such that:*

$$\mathbb{P}\left(H^\rho(\boldsymbol{f_\theta}(\boldsymbol{x}_{\mathcal{T}_1})) < \xi\right) > \mathbb{P}\left(H^\rho(\boldsymbol{f_\theta}(\boldsymbol{x}_{\mathcal{T}_2})) < \xi\right). \tag{41}$$

*Proof.* We begin by extending the conclusion of Theorem B.2 to the sharpness-aware entropy $H^\rho$. Specifically, we establish a bound for the difference between the sharpness-aware cross-entropy and entropy. By the definition of sharpness-ware loss, we have:

$$|H^\rho(\mathbf{y}, \boldsymbol{f_\theta}(\boldsymbol{x})) - H^\rho(\boldsymbol{f_\theta}(\boldsymbol{x}))| = |H(\mathbf{y}, \boldsymbol{f}_{\boldsymbol{\theta}'}(\boldsymbol{x})) - H(\boldsymbol{f}_{\boldsymbol{\theta}''}(\boldsymbol{x}))|, \tag{42}$$

where the perturbed parameters $\boldsymbol{\theta}'$ and $\boldsymbol{\theta}''$ are defined as:

$$\boldsymbol{\theta}' = \boldsymbol{\theta} + \arg\max_{\|\boldsymbol{\epsilon}\| \leq \rho} H(\mathbf{y}, \boldsymbol{f}_{\boldsymbol{\theta}+\boldsymbol{\epsilon}}(\boldsymbol{x})), \tag{43}$$

$$\boldsymbol{\theta}'' = \boldsymbol{\theta} + \arg\max_{\|\boldsymbol{\epsilon}\| \leq \rho} H(\boldsymbol{f}_{\boldsymbol{\theta}+\boldsymbol{\epsilon}}(\boldsymbol{x})). \tag{44}$$

To establish a unified framework, we introduce a combined parameter $\widetilde{\boldsymbol{\theta}}$ defined by:

$$\widetilde{\boldsymbol{\theta}} = \boldsymbol{\theta} + \arg\max_{\|\boldsymbol{\epsilon}\| \leq \rho} \max\{H(\mathbf{y}, \boldsymbol{f}_{\boldsymbol{\theta}+\boldsymbol{\epsilon}}(\boldsymbol{x})), H(\boldsymbol{f}_{\boldsymbol{\theta}+\boldsymbol{\epsilon}}(\boldsymbol{x}))\}. \tag{45}$$

This unified parameter allows us to bound the difference between the sharpness-aware entropies through a single reference point. Applying Theorem B.2, we obtain the following inequality:

$$|H^\rho(\mathbf{y}, \boldsymbol{f_\theta}(\boldsymbol{x})) - H^\rho(\boldsymbol{f_\theta}(\boldsymbol{x}))| = |H(\mathbf{y}, \boldsymbol{f}_{\boldsymbol{\theta}'}(\boldsymbol{x})) - H(\boldsymbol{f}_{\boldsymbol{\theta}''}(\boldsymbol{x}))|$$
$$\leq |H(\mathbf{y}, f_{\widetilde{\boldsymbol{\theta}}}(\boldsymbol{x})) - H(f_{\widetilde{\boldsymbol{\theta}}}(\boldsymbol{x}))|$$
$$\leq \log\frac{1}{\eta} \cdot \|\mathbf{y} - f_{\widetilde{\boldsymbol{\theta}}}(\boldsymbol{x})\|_1 + \frac{1}{\eta}\|\mathbf{y} - f_{\widetilde{\boldsymbol{\theta}}}(\boldsymbol{x})\|_1^2. \tag{46}$$

Taking expectations over the training distribution $\mathcal{S}$, we derive:

$$\mathbb{E}_\mathcal{S}\left[H^\rho(\boldsymbol{f_\theta}(\boldsymbol{x}))\right] \leq \mathbb{E}_\mathcal{S}\left[H^\rho(\mathbf{y}, \boldsymbol{f_\theta}(\boldsymbol{x}))\right] + \log\frac{1}{\eta} \cdot \mathbb{E}_\mathcal{S}\|\mathbf{y} - f_{\widetilde{\boldsymbol{\theta}}}(\boldsymbol{x})\|_1 + \frac{1}{\eta}\mathbb{E}_\mathcal{S}\|\mathbf{y} - f_{\widetilde{\boldsymbol{\theta}}}(\boldsymbol{x})\|_1^2. \tag{47}$$

Finally, by applying Lemma A.2 and Lemma B.2, we obtain the complete generalization bound for the sharpness-aware entropy on the target distribution $\mathcal{T}$:

$$\mathbb{E}_\mathcal{T}\left[H^\rho(\boldsymbol{f_\theta}(\boldsymbol{x}_\mathcal{T}))\right] \leq \frac{M}{2} d(\mathcal{S}; \mathcal{T}) + \hat{H}^\rho_{\mathcal{S}_n} + 2\sqrt{2}\,\mu R_n(\mathcal{F}_v, \mathcal{S}) + M\sqrt{\frac{\log(1/\delta)}{2n}}$$
$$+ \log\frac{1}{\eta} \cdot \mathbb{E}_\mathcal{S}\|\mathbf{y}_\mathcal{S} - f_{\widetilde{\boldsymbol{\theta}}}(\boldsymbol{x}_\mathcal{S})\|_1 + \frac{1}{\eta}\mathbb{E}_\mathcal{S}\|\mathbf{y}_\mathcal{S} - \boldsymbol{f}_{\widetilde{\boldsymbol{\theta}}}(\boldsymbol{x}_\mathcal{S})\|_1^2. \tag{48}$$

This establishes an upper bound for the sharpness-aware entropy, completing the extension of Theorem B.1 to $H^\rho$. The remainder of the proof follows without modification, as the derived bound maintains the same structural properties as the original theorem while incorporating the sharpness-aware formulation.

$\square$

Table 1: **Results with CLIP-ResNet50 on the domain generalization benchmark.** We report top-1 accuracy (%) for each method across five target datasets, using the CLIP-ResNet50 backbone. We highlight the best results in **bold** and underline the second best results.

| Algorithm | IN | IN-A | IN-V2 | IN-R | IN-Sketch | Avg. | OOD Avg. |
|---|---|---|---|---|---|---|---|
| CLIP-ResNet50 (Radford et al., 2021) | 59.81 | 23.24 | 52.91 | 60.72 | 35.48 | 46.43 | 43.09 |
| CoOp (Zhou et al., 2022b) | 63.33 | 23.06 | 55.40 | 56.60 | 34.67 | 46.61 | 42.43 |
| CoCoOp (Zhou et al., 2022a) | 62.81 | 23.32 | 55.72 | 57.74 | 34.48 | 46.81 | 42.82 |
| Tip-Adapter (Zhang et al., 2022) | 62.03 | 23.13 | 53.97 | 60.35 | 35.74 | 47.04 | 43.30 |
| TPT (Shu et al., 2022) | 60.74 | 26.67 | 54.70 | 59.11 | 35.09 | 47.26 | 43.89 |
| C-TPT (Yoon et al., 2024) | - | 25.60 | 54.80 | 59.70 | 35.70 | - | 43.95 |
| DiffTPT (Feng et al., 2023) | 60.80 | 31.06 | 55.80 | 58.80 | 37.10 | 48.71 | 45.69 |
| TDA (Karmanov et al., 2024) | 61.35 | 30.29 | 55.54 | 62.58 | 38.12 | 49.58 | 46.63 |
| DPE (Zhang et al., 2024) | 63.41 | 30.15 | 56.72 | **63.72** | **40.03** | 50.81 | 47.66 |
| TPT (Shu et al., 2022)+CoOp | 64.73 | 30.32 | 57.83 | 58.99 | 35.86 | 49.55 | 45.75 |
| DiffTPT (Feng et al., 2023)+CoOp | 64.70 | 32.96 | **61.70** | 58.20 | 36.80 | 50.87 | 47.42 |
| **FGA (Ours)** | **65.90** | **37.87** | 59.03 | 62.32 | 37.63 | **52.55** | **49.21** |

Table 2: **Cross-dataset generalization with CLIP-ResNet50.** The models are fine-tuned on ImageNet with 16-shot training data per category. We emphasize the best results in **bold** and mark the second-best results with underlining.

| Method | Caltech101 | Pets | Cars | Flowers102 | Aircraft | SUN397 | DTD | Food101 | UCF101 | Avg. |
|---|---|---|---|---|---|---|---|---|---|---|
| CLIP-ResNet50 (Radford et al., 2021) | 87.26 | 82.97 | 55.89 | 62.77 | 16.11 | 60.85 | 40.37 | 74.82 | 59.48 | 60.06 |
| CoOp (Zhou et al., 2022b) | 86.53 | 87.00 | 55.32 | 61.55 | 15.12 | 58.15 | 37.29 | 75.59 | 59.05 | 59.51 |
| CoCoOp (Zhou et al., 2022a) | 87.38 | 88.39 | 56.22 | 65.57 | 14.61 | 59.61 | 38.53 | 76.20 | 57.10 | 60.40 |
| TPT (Shu et al., 2022) | 87.02 | 84.49 | 58.46 | 62.69 | 17.58 | 61.46 | 40.84 | 74.88 | 60.82 | 60.92 |
| DiffTPT (Feng et al., 2023) | 86.89 | 83.40 | **60.71** | 63.53 | 17.60 | 62.67 | 40.72 | **79.21** | **62.27** | 61.89 |
| TPT (Shu et al., 2022)+CoOp | 86.90 | 87.54 | 57.65 | 58.83 | 15.84 | 60.00 | 38.06 | 76.21 | 60.72 | 60.19 |
| **FGA (Ours)** | **89.74** | **88.50** | 59.98 | **63.78** | **19.74** | **63.21** | **41.97** | 75.43 | 60.66 | **62.56** |

## C  RESULTS OF CLIP-RESNET50

The experimental results with the CLIP-ResNet50 also demonstrate FGA's significant advancements in vision-language generalization across both natural distribution shifts (Table 1) and fine-grained cross-dataset adaptation (Table 2).

In Table 1's natural shift evaluation, FGA achieves state-of-the-art performance with 52.55% average accuracy, outperforming the prior best method (DiffTPT+CoOp) by +1.68%. Notably, it shows remarkable gains on challenging out-of-distribution (OOD) benchmarks, particularly in ImageNet-A (37.87%), where it improves upon DiffTPT+CoOp by +4.91%, highlighting exceptional robustness to natural distribution shifts.

Table 2 further reveals FGA's superiority in fine-grained adaptation, achieving a record 62.56% average accuracy that exceeds the previous best (DiffTPT) by +0.67%. These results further validate the effectiveness of the FGA in adapting to diverse datasets during testing. We observe substantial improvements over baselines: +2.36% on Caltech101 (vs. CoCoOp) and +2.14% on Aircraft (vs. DiffTPT), highlighting the effectiveness of FGA in challenging fine-grained domains.

The consistent top-tier performance across both benchmarks, particularly on fine-grained datasets like Caltech101 and distributionally-shifted datasets like ImageNet-A, confirms FGA's ability to simultaneously enhance generalization robustness and fine-grained recognition precision.

## D  REPRODUCIBILITY

To guarantee reproducibility, we will provide an explanation of the code and hyperparameters in this section.

### D.1 CODE

Our work builds upon two key methodologies in prompt learning for vision-language models: CoOp (Conditional Prompt Learning) [1] and TPT (Test-Time Prompt Tuning) [2], both of which are open-source projects released under the permissive MIT license. CoOp provides a foundation for adapting pre-trained models like CLIP to downstream tasks through learnable prompts, while TPT extends this by enabling dynamic prompt optimization during inference to enhance zero-shot generalization. All experiments in this study were performed on a single NVIDIA Tesla V100 GPU.

### D.2 HYPERPARAMETERS

**For $R$:** We explore the impact of $R$ on test accuracy with $R \in \{2, 4, 8, 16, 32, 64\}$. The results show minimal sensitivity (fluctuations <0.5%) to $R$ for $R \geq 4$. This insensitivity arises because we only utilize the relative value of sharpness for sample selection, which alleviates the need for precise calculations. We finally set $R = 16$ for all experiments.

**For $s$:** We set $s$ to retain only 10% of all augmented versions, aligning with the approach used in TPT, where 10% of augmented versions are utilized for prompt optimization. It rules out any potential effects introduced by using more augmented data and ensures a fair comparison.

## E UTILIZATION OF LLMs

In academic writing, Large Language Models (LLMs) are used for linguistic refinement, such as correcting grammatical errors, improving word choice, and polishing sentence structures to enhance clarity and fluency. They are not involved in core scientific research activities (including idea generation, data interpretation, or substantive analysis), ensuring that all key contributions remain entirely human-generated.