# OpenReview forum: "Flatness Guided Test-Time Adaptation for Vision-Language Models"
_ICLR.cc/2026/Conference — ICLR 2026 Poster_

### Official Review · Reviewer_BBAK · 2025-10-15

**Soundness:** 2
**Presentation:** 3
**Contribution:** 3
**Rating:** 6
**Confidence:** 3

**Summary:**

Most of existing test-time adaption methods isolate the test phase adaption from the training phase, leading to sub-optimal generalization performance. To mitigate this gap, this paper leverages the alignment between the training minimum and test loss flat regions to guide the test-time adaptation process, and proposes a flatness-guided adaptation framework (FGA). The effectiveness of FGA is verified both by theoretical analyses and extensive experiments.

**Strengths:**

* This paper is well-written and easy to follow.

* The proposed framework FGA is novelty and well-motivated. The experimental results also demonstrate its effectiveness.

* The experimental results are sufficient.

**Weaknesses:**

There are some concerns on the theoretical analyses.

* The generalization error bound in Theorem 1 is not tight and its effectiveness can be very restricted.

In Theorem 1, there is a term $M\sqrt{\frac{1}{2}\cdot\ln\frac{2}{\delta}}$ on the upper bound. It seems that this generalization error bound is efficient only for $\delta>\frac{2}{\mathrm{e}^2}$. Otherwise, this upper bound is larger than $M$, a trivial result.

* In Theorem 4, there is a lack of an upper bound on $\xi$. If $\xi$ is very large, then it is hard to distinct two test distributions by (12).

**Questions:**

* In Theorem 1, what are the definitions of $d_{F\Delta F}$ and $\hat{\ell}^{\rho}$ ? Does the loss function in the theorem i.e., $\ell^p()$, equal to $\ell_{STSS}$ ?


* In (5), why are the perturbations sampled from the standard normal distribution? Is it used to approximate the direction of gradient (similar to (4))? If so, why is such an approximation strategy reasonable?

* In Theorem 4, what is the relation between $\xi$ and $\beta$ ? In Theorem 1, if $\delta>\frac{2}{\mathrm{e}^2}$, then there is a trivial upper bound on the generalization, i.e., $M$. By Definition 2, we can conclude $\beta=0$. In this case, is $\xi=M$ ?

* FGA is independent of the methods that augment test samples. In the experiment, how does FGA augment test samples?

---

> ### Author Response · Authors · 2025-11-22
>
> Dear Reviewer BBAK,
>
> Thank you for your thorough and insightful review. We sincerely appreciate the time and effort you dedicated to evaluating our work. Below, we provide a point-by-point response to each of the comments and concerns you raised.
>
> **Q4-1:** **About the definition of  $d_{\mathcal{F}\Delta\mathcal{F}}$, $\ell^\rho$ and $\ell_{STSS}$.**
>
> **A4-1:** The definition of the discrepancy measure $d_{\mathcal{F}\Delta\mathcal{F}}$ is provided in Appendix A; The notation $\widehat{\ell}^{\rho}$ denotes the empirical counterpart of the loss function $\ell^\rho$, following the common convention in the machine learning community. We will include necessary explanations on these points in the final version of our paper. Additonally, $\ell_{STSS}$ acts as an efficient approximation of $\ell^\rho$ in the absence of test labels, with the utilization of a surrogate loss and an approximation of worst-case perturbation direction. We provide a detailed analysis of this approximation in **Q4-2**.
>
> **Q4-2:** **About Approximation in Equation (5).**
>
> **A4-2:** In Equation (5), to avoid the expensive gradient calculation in Equation (4), we approximate the worst-case perturbation direction by selecting the most adversarial direction from $M$ randomly sampled vectors. These perturbation vectors are sampled from a standard normal distribution. Because the standard normal distribution is isotropic, the normalized perturbations (i.e., the perturbation directions) follow a uniform distribution on the unit sphere, ensuring that every direction has an equal probability of being sampled.
>
> Intuitively, one might expect that a large value of $M$ would be necessary to obtain a highly accurate approximation. However, our selection mechanism relies solely on the relative ranking of the values. This relaxes the requirement for high-precision approximation, allowing us to achieve effective performance with a relatively small value of $M$.
>
> Importantly, although Equation (5) employs the aforementioned approximation, our theoretical analysis remains valid and meaningful. The introduced approximations can be equivalently interpreted as introducing an additional error term into the generalization bound. This interpretation leads to a  bound that is $\beta'$-tight, where $\beta'> \beta$. An implication of this, according to Theorem 2, is that a larger separation between two distributions may be required to reliably distinguish them. However, this requirement does not fundamentally invalidate our analysis, as it merely quantifiably relaxes the bound's tightness. We have explicitly mentioned this point in Remark 1 of Appendix B. Providing a rigorous analysis is extremely challenging, and still remains a valuable direction for our future work.
>
> **Q4-3:** **Concern on Theoretical Analysis.**
>
> **A4-3:** Thank you for pointing out the issue of the trivial upper bound that arises when the confidence parameter $\delta$ is less than $\frac{2}{e^2}$.
>
> It is crucial to clarify that the bound in equation (11) specifically applies to the loss of a single sample drawn from the test distribution $\mathcal{T}$, denoted as $\ell^{\rho}(f(X_{\mathcal{T}}), Y_{\mathcal{T}})$ (abbreviated as $\ell^{\rho}$ hereafter), rather than to its expectation over $\mathcal{T}$. To characterize the deviation of $\ell^{\rho}$ from its expected value, we rely on an upper bound for the tail distribution of bounded random variables, which yields the term $M\sqrt{\frac{\log(2/\delta)}{2}}$ by using Hoeffding's inequality. Unfortunately, this trivial interval (where the bound exceeds the maximum loss $M$) is nearly unavoidable in similar bounds due to the inherent boundedness of $\ell^{\rho}$.
>
> Importantly, this limitation does not undermine the correctness of Theorem 2. If a tighter bound becomes available, it can be directly substituted, thereby reducing the range of $\delta$ for which the bound becomes trivial. To effectively address the issue, we can assume an ideal upper bound, $F(\delta)$, defined such that $\mathrm{Pr}(\ell^{\rho} \leq \mathbb{E}[\ell^{\rho}] + F(\delta)) = 1 - \delta$, which corresponds to the quantile function of the loss distribution. In this ideal upper bound, the issue of a trivial interval will disappear.
>
> **Q4-4:** **About Data Augmentation.**
>
> To rule out the benefit brought by data augmentation and ensure a fair comparison with other methods, we adopt the same data augmentations as TPT, including random crops, flip and AugMix. Even using these simple data augmentations, our FGA already exhibits superior performances over existing methods. Developing more advanced data augmentations to handle more challenging cases remains a valuable future research.
>
> Thanks again for your detailed review and feedback. We will incorporate valuable details into the latest version later.

---

> > ### Comment · Reviewer_BBAK · 2025-11-26
> >
> > I thank the authors for detailed rebuttal that have addressed most of my concerns.
> > The first weakness remains unchanged.
> >
> > To dervive a high-probability generalization error bound,
> > it is necessary to use some type of concentration inequality, such as the Hoeffding's inequality.
> > Even if an expected generalization bound is derived,
> > the second term in the upper bound would be a constant term that depends on $M$.
> > I conjecture that the second term in the upper bound should decay with $n$.

---

> ### Author Response · Authors · 2025-11-26
> **About further discussion about generalization error bound**
>
> Dear Reviewer BBAK,
>
> Thanks for your further discussion that the term $M\sqrt{\frac{\log(2/\delta)}{2}}$ in the upper bound would be a constant term that does not decay with $n$.
>
> We would like to clarify a fundamental distinction in the scope of our analysis. Typical generalization bounds in statistical learning theory most commonly analyze the **expected loss**, denoted as $\mathbb{E}[\ell^\rho]$, over the entire test distribution $\mathcal{T}_i$ . In contrast, the bound presented in our Equation (11) is specifically derived for the loss $\ell^\rho$ of a **single sample** drawn from $\mathcal{T}_i$.
>
> This distinction is critical. For the expected loss $\mathbb{E}[\ell^\rho]$, which represents an average over the entire data distribution, it is indeed possible to derive bounds with terms that decay to zero as the number of training samples $ n $ increases . However, for the loss on a single sample, it is inherently impossible for all terms in the bound to decay to zero. To see why, consider that even with an infinite number of training samples, there might always exist some individual test sample whose loss is inherently large, leading to an incorrect prediction. If all terms in the single-sample bound could vanish, it would imply that the model could perfectly predict every single test point, achieving 100% test accuracy, which is generally unattainable.
>
> Additionally, it is reasonable that our theoretical conclusion may not hold for a few potentially "bad" test samples. This is a natural and reasonable characteristic of single-sample conclusions, as guaranteeing correctness for every single possible sample is an unrealistic expectation.
>
> By the way, Theorems 3.1 and 3.4 have been revised to further strengthen the theoretical foundation of our core claims.
>
> I hope this answers your question well. Thanks again!

---

### Official Review · Reviewer_P1eB · 2025-10-26

**Soundness:** 3
**Presentation:** 3
**Contribution:** 3
**Rating:** 6
**Confidence:** 3

**Summary:**

This paper addresses test-time adaptation (TTA) for vision-language models (VLMs) such as CLIP. Test-time prompt tuning (TPT), which optimizes text prompts corresponding to classes, is the mainstream approach in this field. TPT optimizes prompts for test samples using surrogate losses like entropy minimization. Since this optimization relies on backpropagation, it incurs significant overhead and substantially increases inference time. To this end, this paper proposes a novel test-time adaptation paradigm called flatness-guided adaptation (FGA). The core idea of FGA consists of two parts: adding regularization during the prompt-tuning phase to flatten the loss landscape, and selecting data augmentation samples during inference based on sharpness scores from the surrogate loss. This enables adapting the model to the test distribution without backpropagation during testing. Theoretically, the paper demonstrates that selecting similar samples via sharpness scores can bring the test distribution closer to the training distribution, potentially improving generalization performance. Experimentally, across standard benchmarks, the paper shows that FGA significantly improves performance compared to existing TPT baselines while substantially reducing inference time.

**Strengths:**

- **S1.** The paper proposes a novel TTA paradigm that eliminates the need for backpropagation during testing. This is innovative.
- **S2.** The paper theoretically discusses the relationship between sharpness and generalization performance on the loss landscape, justifying the effectiveness of the proposed method. If this analysis is novel, it is expected to have a significant impact on the field.
- **S3.** The paper evaluates the performance and inference speed improvements achieved by the proposed method through experiments using standard benchmarks. In particular, the comparison with the baseline combining CoOp and TPT is carefully designed and fair.

**Weaknesses:**

- **W1.** Section 4's theoretical analysis discusses generalization performance using the loss $l^\rho$ (e.g., cross-entropy for classification), but the proposed method shown in Section 3 actually evaluates sharpness using the surrogate loss $l_\text{SRG}$, causing a discrepancy with reality and theory. The paper should add theoretical supplements describing the effects by this gap. Furthermore, practical insights such as the correlation between sharpness computed via cross-entropy using ground-truth labels and sharpness computed via entropy would also be desirable.
- **W2.** FGA can indeed avoid backpropagation during inference, but the additional training cost due to flatness regularization is significant. While the paper introduces a reasonable approximation calculation via Taylor expansion in Eq. (4), a quantitative discussion is needed on how much training time increases compared to existing prompt-tuning, such as CoOp.
- **W3.** FGA relies on a training phase with teacher labels. This is a significant practical constraint. The paper primarily trains using 16-shot ImageNet, implying a requirement for 16,000 training samples. Collecting datasets of this scale is a major burden in contexts like medical image processing. In this sense, demonstrating experimental results with fewer shots would significantly strengthen the paper's claims and enhance its value for practitioners, though it already shows "zero-shot" performance as "FGA wo/ SAPT" in Table 1.

**Questions:**

Please respond to the concerns raised in the weaknesses section.

---

> ### Author Response · Authors · 2025-11-22
>
> Dear Reviewer P1eB,
>
> Thank you for your thorough and insightful review. We sincerely appreciate the time and effort you dedicated to evaluating our work. Below, we provide a point-by-point response to each of the comments and concerns you raised.
>
> **Q3-1:** **Concern on Empirical-Theoretical Gap in Surrogate Losses**
>
> **A3-1:** Although Equation (5) employs a surrogate loss rather than the true cross-entropy loss, our theoretical analysis remains valid and meaningful. The introduced approximations can be equivalently interpreted as introducing an additional error term into the generalization bound. This interpretation leads to a  bound that is $\beta'$-tight, where $\beta'> \beta$. An implication of this, according to Theorem 2, is that a larger separation between two distributions may be required to reliably distinguish them. However, this requirement does not fundamentally invalidate our analysis, as it merely quantifiably relaxes the bound's tightness. We have explicitly metioned this point in Remark 1 of Appendix B. Providing a rigorous analysis is extremely challenging, and still remains a valuable direction for our future work.
>
> **Q3-2:** **About Training time compared to CoOp.**
>
> **A3-2:** Thanks for your concern on the increased training time associated with flatness regularization. In fact, given the well-established correlation between flatness and generalization, flatness-related regularization has been extensively studied in recent years.
> Although SAM-inspired methods often incur additional computational overhead for approximating the sharpness measure, several methods have been specially developed to mitigate this overhead. For example, SAF[1] can mitigate the sharp at almost zero additional computational cost. Integrating such efficient SAM variants into our FGA framework is straightforward, and we plan to implement this in future work.
>
> *[1] Du, Jiawei, et al. "Sharpness-aware training for free." Advances in Neural Information Processing Systems 35 (2022): 23439-23451.*
>
> **Q3-3:** **About Prompt Tuning with Fewer Training Shots.**
>
> **A3-3:** Thank you for your comment regarding additional experiments with fewer shots during prompt tuning. In our current experimental setup, we have followed the established protocol from the seminal TPT paper, where CoOp is tuned on ImageNet using 16 training shots per category. This configuration has also been consistently adopted by a series of subsequent TTA methods built upon TPT. To maintain a fair and consistent basis for comparison with these existing approaches, we adhered to the same experimental setting. We will certainly consider your suggestion for exploring fewer-shot scenarios in future extensions of this work.
>
> Thanks again for your detailed review and feedback. We will incorporate valuable details into the latest version later.

---

> > ### Comment · Reviewer_P1eB · 2025-11-27
> >
> > Thank you for your response.
> >
> > Regarding Q3-1, I understand that the bound changes from $\beta$ to $\beta'$. The tightness of this bound by $\beta'$ is intriguing, so a detailed analysis is anticipated.
> >
> > Regarding Q3-2, the response mentioned existing research and future directions, but it did not present concrete evidence in this paper's case, nor did it directly address my concern.
> >
> > Regarding Q3-3, I understand the experimental setup follows standard protocols. However, this does not resolve my concern, i.e., the performance trend of the proposed method when fewer shots are used.
> >
> > Overall, while the responses partially addressed my concerns, the provided evidence is somewhat weak. I understand that additional experiments are optional and should not be demanded by reviewers, so I would like to discuss and determine the final recommendation score with the other reviewers.
> >
> > Thank you again for your thoughtful responses.

---

> > > ### Author Response · Authors · 2025-11-30
> > >
> > > Thank you for the further comments. We appreciate the opportunity to further clarify these points. Below, we provide a point-by-point response to the specific concerns raised.
> > >
> > > *   **Regarding Q3-1:** As suggested, we have provided additional theoretical analysis in the revised manuscript (see Section 4) to analyse the correlation between the sharpness-aware cross-entropy and the sharpness-aware entropy. This revision further strengthens the theoretical foundation of our core claims.
> > >
> > > *   **Regarding Q3-2:** We thank the reviewer for this comment. It is crucial to clarify that the primary aim of our work is not to propose a new algorithm for finding flat minima, which represents an area already well-explored in the literature. Instead, our central contribution lies in  demonstrating that the geometric property of flatness, acquired during training, can directly guide test-time adaptation. Within our proposed FGA framework, any efficient flat-minima seeking method can be incorporated for fast prompt-based tuning. A comprehensive comparison of the computational efficiency of various seekers, while important, falls within the scope of other specialized research lines.
> > >
> > > *   **Regarding Q3-3:** We agree that exploring broader applications is valuable. However, adopting the well-established experimental protocol allowed us to focus on rigorously validating the core effectiveness of FGA. While we acknowledge the potential of FGA for broader use, a systematic extension to other experimental configurations would require substantial engineering efforts to conduct large-scale experiments. We have incorporated this important direction into the "Future Work" section of our revised manuscript.
> > >
> > > We believe these revisions and clarifications could address your concerns well. Thanks once again for your further detailed feedback.

---

### Official Review · Reviewer_j4NZ · 2025-10-29

**Soundness:** 2
**Presentation:** 3
**Contribution:** 2
**Rating:** 4
**Confidence:** 5

**Summary:**

This paper proposes Flatness-Guided Adaptation (FGA) for test-time adaptation of CLIP, leveraging sharpness-aware training to align training and test loss landscapes without updating parameters at test time.It introduces SAPT and STSS to guide adaptation using flatness as a geometric clue, enhancing generalization.

**Strengths:**

1. The paper introduces a FGA method that cohesively links training and test-time adaptation through the concept of loss landscape flatness, improving robustness to distribution shifts without updating model parameters during inference.
2. FGA avoids backpropagation and prompt updates at test time, resulting in lower computational overhead while still improving performance.

**Weaknesses:**

1. The scope of the paper is mismatched: While the title refers to test-time adaptation (TTA) for ``vision-language models" broadly, all experiments are limited to CLIP variants (ViT-B/16 and ResNet50). This narrow scope excludes newer or structurally different VLMs such as  LLAVA or CLIP variants such as SigLIP, SigLIP-v2, or other similar models, undermining the generality claim.
2. The phrase ``training data on downstream task" is vague. From the experimental section, it seems the prompt tuning is conducted on ImageNet, while evaluation is on ImageNet variants. If ImageNet is treated as the training data, this may conflict with standard TTA setups, where benchmark dataset ImageNet should be unseen during training to preserve test-time integrity.
3. The method combines sharpness-aware prompt tuning (SAPT) and test-time selection based on sharpness. While effective, this formulation closely relates to ideas explored in prior work like DPO [1], which also applies perturbation-based optimization for TTA, albeit in 3D object detection. A discussion of its differences/similarities would strengthen the positioning of this work.
4. The experimental comparisons are inconsistent. While Table 1 includes TDA, Table 2 omits it despite being a relevant baseline for cross-dataset generalization.
5. Recent methods like TCA [2] report higher performance than FGA without requiring augmentations or training overhead. The absence of direct comparison with such state-of-the-art approaches weakens the empirical claims.

[1] DPO: Dual-Perturbation Optimization for Test-time Adaptation in 3D Object Detection

[2] Is Less More? Exploring Token Condensation as Training-free Test-time Adaptation

**Questions:**

See the Weaknesses

---

> ### Author Response · Authors · 2025-11-22
>
> Dear Reviewer j4NZ,
>
> We sincerely appreciate the time and effort you dedicated to evaluating our work. Below, we provide a point-by-point response to each of the concerns you raised.
>
> **Q2-1:** **Concern about the scope of this paper.**
>
> **A2-1:** Thanks for this insightful comment about the scope of our work. It is true that our current work focuses on CLIP-based architectures, which was a deliberate choice given their established role as standard benchmarks in the vision-language community. This approach follows common practice in the test-time adaptation literature, as seen in prior works such as TPT, TDA, and DPE, which also concentrate on CLIP-based models despite the broader scope implied by their titles.
>
> Importantly, our FGA framework is model-agnostic, relying only on general components such as prompt-based conditioning and loss landscape geometry, which are shared across many VLMs. We plan to include more experiments in the future to further demonstrate the broad applicability of FGA.
>
> **Q2-2:** **Concern on Violations of TTA Setups.**
>
> **A2-2:** For more clarity, a formal definition of TTA is presented below. TTA methods can be categorized as two main branches: those that modify the training process (training-preparation ones) and those that adapt models without altering the training process (preparation-agnostic ones). By the definition of TTA, "training data on downstream task" corresponds to the source dataset. It is important to note that while some influential preparation-agnostic methods (e.g., TPT) do not utilize the source dataset to evaluate zero-shot generalization, it does not imply that using it is prohibited in TTA setups. For example, TPT+CoOp presented in the TPT paper, explicitly involves a prompt tuning process on the source dataset.
>
> **Definition 1(Test-time adaptation [1]).** Given labeled source distributions $\mathcal{S}$ and unlabeled target distributions $\mathcal{T}$, test-time adaptation aims to train a model $f_{\theta_{s}}$ only on the source distribution $\mathcal{S}$, and achieve adaptation using the source-trained model $f_{\theta_{s}}$ and the target data $x_{t}$ to make predictions on $x_{t}$ after adaptation.
>
> *[1] Xiao Z, Snoek C G M. Beyond model adaptation at test time: A survey[J]. arXiv preprint arXiv:2411.03687, 2024.*
>
> **Q2-3:** **Comparisons with DPO.**
>
> **A2-3:** While both DPO and our FGA leverage the concept of sharpness-awareness, they are totally different in both core idea and methodology.
>
> From the core idea, DPO actively searches for a flat minimum within the test loss landscape during testing. In contrast,  FGA leverages the alignment between this pre-acquired training flat minimum and the test flat regions to guide the adaptation process. Instead of optimizing for a new minimum at test time, FGA keeps model parameters frozen. We think Figure 1 can clearly demonstrate this difference.
>
> From the methodology, (1) At test time, they use different methods to evaluate loss sharpness: DPO uses a gradient-based method, while FGA injects random perturbations to model parameters; (2) Their input space is changed differently: DPO introduces expensive adversarial perturbations to the input space, aiming to better simulate the noisy test environment. In contrast, FGA introduces data augmentations to the input space, aiming to alter the loss landscape for the alignment of flat minima.
>
> **Q2-4:** **Comparisons with TDA and TCA.**
>
> **A2-4:** We present a detailed comparison in the following table. Notably, FGA is specifically designed for the challenging single-sample​ scenario, different from online TTA methods that benefit from aggregated information across a continuous data stream. Thus, online TTA methods (e.g., TCA) may exhibit extremely high performance (70.43%) on certain datasets (e.g., EuroSAT). It is crucial to note that a direct comparison between FGA and these online TTA methods is inherently unfair. Despite this inherent disadvantage, FGA demonstrates remarkable effectiveness by outperforming methods like TDA and TCA (R=0.7) on overall benchmark accuracy.
>
> | Method | Caltech101 | Pets | Cars | Flowers102 | Aircraft | SUN397 | DTD | EuroSAT | Food101 | UCF101 | Avg |
> | :--- | :--- | :--- | :--- | :--- | :--- | :--- | :--- | :--- | :--- | :--- | :--- |
> | CLIP-ViT-B/16 | 93.35 | 88.25 | 65.48 | 67.44 | 23.67 | 62.59 | 44.27 | 42.01 | 83.65 | 65.13 | 63.58 |
> | **TDA** | 94.24 | 88.63 | 67.28 | 71.42 | 23.91 | 67.54 | 47.40 | 58.00 | **86.14** | 70.66 | 67.53 |
> | **TCA(R=0.7)** | 92.13 | 85.99 | 58.15 | 69.79 | 23.19 | 61.89 | 44.50 | 61.63 | 79.99 | 67.38 | 64.46 |
> | **TCA(R=0.9)** | 93.63 | 89.53 | 65.33 | **73.33** | 24.87 | 65.92 | 46.16 | **70.43** | 85.31 | **72.38** | **68.69** |
> | **FGA** | **96.96** | **91.28** | **68.93** | 72.11 | **26.97** | **69.29** | **49.76** | 47.58 | 84.95 | 68.17 | 67.60 |
>
> Thanks again for your detailed review and feedback. We will incorporate valuable details into the latest version later.

---

> > ### Comment · Reviewer_j4NZ · 2025-11-25
> >
> > Could you explain more about "It is crucial to note that a direct comparison between FGA and these online TTA methods is inherently unfair. " in your reply? Thanks

---

> ### Author Response · Authors · 2025-11-25
> **About  unfairness in comparing FGA with online TTA methods**
>
> Dear Reviewer j4NZ,
>
> Thanks for your valuable feedback, which provides an opportunity for further discussion. The unfairness in comparing FGA with online TTA methods fundamentally arises from their belonging to distinct categories of Test-Time Adaptation.
>
> We can clarify this by examining two primary strategies in TTA: **online TTA** and **per-sample isolated adaptation**. The core distinction lies in how historical test data is utilized during adaptation, which directly impacts the fairness of comparison. Our discussion below is based on a model initialized with parameters $\theta_0$ and a sequence of test samples $(x_1, x_2, \ldots, x_n)$, considering methods that update model parameters for illustration.
>
> -   The **online TTA** strategy updates model parameters sequentially as test samples arrive: $\theta_0 \rightarrow \theta_1 \rightarrow \cdots \rightarrow \theta_n$. When processing the $n$-th sample $x_n$, the model starts from parameters $\theta_{n-1}$, which have integrated information from all preceding samples $(x_1$ to $x_{n-1})$. This allows the model to leverage aggregated historical data for predictions, making it suitable for streaming data scenarios where distribution shifts gradually.
> -   In contrast, the **per-sample isolated adaptation** strategy adapts the model individually for each test sample, always starting from the original parameters $\theta_0$. The update path is $\theta_0 \rightarrow \theta_i \rightarrow \theta_0$ for each sample $x_i$. Notably, the model's state is reset to  $\theta_0$ before processing the next sample. This ensures each adaptation is independent and unaffected by previous test samples, making it more appropriate for scenarios with scarce or non-stationary test data where historical information is unreliable or unavailable.
>
> It is important to note that although recent TTA methods do not update parameters, online approaches still implicitly incorporate historical test sample information (e.g., a Token reservoir in TCA or  key-value cache in TDA). In contrast, the FGA method discussed in this paper, operates under a strict setting where no historical information can be leveraged, and adaptation is constrained strictly to individual test samples. This fundamental difference in information access is the root cause of the inherent unfairness in direct comparisons.
>
> Thanks again for raising this important point.

---

> > ### Comment · Reviewer_j4NZ · 2025-11-27
> >
> > Thanks for your reply. From my understanding, all of these are CLIP-based TTA methods aiming to mitigate shift when CLIP models or similar VLMs are applied to downstream tasks.
> >
> > For example, TDA has no trainable parameters, while TCA uses no augmentations and also requires no parameter updates even with lower FLOPs. Still, they should be compared together since they share the same motivation, even though they take different approaches.

---

> ### Author Response · Authors · 2025-11-30
>
> Thanks for the reviewer's further comment.
>
> We would like to clarify that although the motivation behind all these TTA methods is to address domain shift, **they differ fundamentally in how much target domain information is utilized during adaptation**. Specifically, online TTA methods (TDA and TCA) are allowed to leverage a greater amount of target domain information, which is aggregated from historical test samples streamed sequentially. This fundamental setting makes online TTA methods infeasible under the per-sample TTA setting, as the temporal information aggregation cannot be obtained from a single test sample. In contrast, our FGA method operates under this more challenging per-sample TTA setting, where only the individual test sample is available for adaptation.
>
> Thus, due to this significant difference in the amount of target domain information available to different TTA settings, it is not imperative to expect per-sample TTA methods to surpass online TTA methods. Their performance difference is primarily a reflection of their fundamentally different problem settings and information constraints, rather than solely an indicator of their relative effectiveness.
>
> We believe this reply could address your concern completely. Thanks again!

---

### Official Review · Reviewer_RFcj · 2025-10-30

**Soundness:** 3
**Presentation:** 3
**Contribution:** 2
**Rating:** 4
**Confidence:** 3

**Summary:**

The article proposes an approach for test-time adaptation of vision language models based on flat minima. Specifically, the approach (FGA) is composed of two elements: (i) sharpness-aware prompt tuning (SAPT), which adds a regularizer toward flat minima when tuning prompts in the style of CoOp (Zhou et al. 2022b), (ii) sharpness-based test-time sample selection (STSS), which augments test samples and selects the top augmented views according to the flatness of their loss landscape. Experiments on three benchmarks show the efficacy of the approach.

**Strengths:**

1. To my knowledge, this is the first work tackling TTA (and prompt tuning) for vision-language models from the point of view of flat minima. This perspective on the problem is sound, supported by both findings of previous works (e.g., Niu et al. 2023, Gong et al. 2023) as well as the theoretical analysis of Sec. 4, and the achieved experimental results. The paper confirms the potential of this strategy, proposing variants suitable for the increased representation capabilities of VLMs.

2. The strategy for selecting samples based on their sharpness in the loss landscape (STSS), is an interesting contribution, providing an alternative way to the commonly employed entropy (e.g., Shu et al. 2023) and being the element with the highest impact on the performance (Fig. 3). These showcases that flatness priors can guide the model at multiple levels, both during training for prompt tuning and during inference for data filtering.

**Weaknesses:**

**1.** While FGA is an interesting approach, its two components could be techniques applied independently for prompt-tuning (SAPT) and for sample selection (STSS). Specifically, these two approaches might be treated independently from each other, as SAPT is a strategy for prompt tuning while STSS is a strategy for TTA with sample selection. Thus:\
&nbsp;&nbsp;&nbsp;**1.1**  SAPT could be, in principle, compared with CoOp and related variants on prompt-tuning benchmarks (e.g., base vs novel categories, as in Zhou et al. 2022a). This would show the benefit of flatness-aware prompt tuning with strong empirical evidence against prompt tuning alternatives. Similarly, as the method builds on CoOp, adding the flatness-aware regularizer of SAPT on top of other prompt-tuning approaches based on simple contrastive objectives (e.g., Zhou et al. 2022a, Khattak et al. 2023) would showcase that the advantages of flatness-aware prompt tuning are not restricted to CoOp-based tuning.\
&nbsp;&nbsp;&nbsp;**1.2** On the other hand, STSS could bring benefits to various approaches performing sample selection for TTA (without considering prompt tuning). In fact, selecting augmentations and aggregating their scores might be a variant for a lot of TTA approaches performing sample selection based on alternative criteria (e.g., entropy-based, as in TPT, Shu et al. 2023, DPE, Zhang et al. 2024a, and ZERO, Farina et al. 2024). This would demonstrate that flatness-aware selection can improve a wide range of methods, strengthening the case for its wider applicability.

**2.** Given that the method is based on both offline prompt tuning and TTA, baselines should comprise three categories of methods: (i) TTA-only strategies, (ii) prompt-tuning only strategies, and (iii) the two merged. While this is the case, the selection of methods for the third category is limited, as, beyond PromptAlign (Abdul Samadh et al. 2024), only TPT and DiffTPT are considered as potential TTA strategies applied to prompt tuning approaches, and not in a consistent manner (e.g., Tab. 1 reports both applied to CoOp, Tab. 2 only TPT on top of CoOp and MaPLe). To fully demonstrate the strength of the approach, more competitive baselines should be built using more recent TTA approaches (e.g., C-TPT, ZERO, [b]) coupled with more recent prompt tuning strategies (e.g., [c,d]), reported in all tables. Moreover, following on the previous point, tables could report the results for SAPT and STSS alone (i.e., coupled with CoOp or alternatives), as prompt-tuning and TTA methods, respectively, ensuring a fair comparison beyond merging solutions for the two problems.

**3.** Another set of strong baselines missing are those related to previous methods proposing flatness-aware optimization for online TTA, i.e.,  SAR (Niu et al. 2023) and SoTTA (Gong et al. 2023). It is worth noting that while the problem they address is different (i.e., online TTA with continuous updates), tables already include comparisons with methods employed in this setup (e.g., TDA, Karmanov et al. 2024, and DPE, Zhang et al. 2024a). Moreover, while it is true that they were not designed for VLMs, an adaptation of these approaches (as done in [a] for SAR) would suffice and strengthen the claim that FGA is the TTA strategy using flatness principles that is most suited for VLMs.

**Minors:**

4. Tables 1 and 2 appear far from the pages where they are described (e.g., Tab. 1 is on page 6 but discussed only on page 8). Reporting the table closer would avoid back and forth between pages when reading the manuscript.

5. In the appendix, the name of the method changes to TLLA.

**References:**\
[a] Wang, Zixin, et al. "Is less more? exploring token condensation as training-free test-time adaptation." ICCV 2025.\
[b] Zanella, Maxime, and Ismail Ben Ayed. "On the test-time zero-shot generalization of vision-language models: Do we really need prompt learning?" CVPR 2024.\
[c] Yao, Hantao, Rui Zhang, and Changsheng Xu. "Visual-language prompt tuning with knowledge-guided context optimization." CVPR 2023.\
[d] Roy, Shuvendu, and Ali Etemad. "Consistency-guided Prompt Learning for Vision-Language Models." ICLR 2024.

**Questions:**

1. Is SAPT an effective strategy for improving prompt-tuning with few shots?
2. Is STSS an effective strategy for improving samples selection for TTA?
3. How would the method compare with more recent baselines (even those merging recent TTA and prompt-tuning strategies)?
4. How would the method (and its individual components) compare with adaptations of SAR and SoTTA?

---

> ### Author Response · Authors · 2025-11-22
>
> Dear Reviewer RFcj,
>
> Thank you for your thorough and insightful review. We sincerely appreciate the time and effort you dedicated to evaluating our work. Below, we provide a point-by-point response to each of the comments and concerns you raised.
>
> **Q1-1:** **About Independent Analysis of SAPT and STSS.**
>
> **A1-1:** Although the two components (SAPT and STSS) of FGA could be applied independently. It is worth noting that FGA is a unified framework, where SAPT and STSS are synergistically designed. Its core idea centered on leveraging the alignment between training flat minima and test loss flat regions to guide the adaptation process. Thus, independently applying SAPT or STSS may not fully harness this alignment.
>
> We sincerely appreciate your valuable suggestion about combining SAPT and STSS with the latest methods. In response, we would like to clarify the reason for the absence of such experiments.
>
> * **About appying SAPT to other prompt-tuning approaches.** Applying SAPT to a wider range of prompt-tuning methods is an interesting direction. However, the main focus of this paper is not to re-establish the well-documented connection between flatness and generalization, which has been extensively validated in recent literature. Instead, our core contribution lies in investigating a novel application: how the geometric property of flatness, acquired during training, can directly guide test-time adaptation. Thus, demonstrating SAPT's superiority over other prompt-tuning baselines falls outside the scope of this paper.
>
> * **About incorprating STSS to other TTA approaches.** Since STSS operates by changing the sample selection criterion, it could theoretically be integrated into other frameworks that rely on augmentation selection. However, most existing TTA methods are based on the entropy criterion, leading to the same selection criterion when combined with STSS. Given that non-entropy-based TTA methods are less explored, incorporating STSS into non-entropy-based TTA methods still remains our future work.
>
> **Q1-2:** **Concerning on Baselines Merging TTA and Prompt Tuning.**
>
> **A1-2:** We sincerely appreciate the valuable suggestion to include additional baselines that combine test-time adaptation (TTA) with prompt tuning methods. The corresponding results on the domain generalization benchmark are presented in the table below. Our findings indicate that even the STSS component of FGA already surpasses other baseline methods. Furthermore, the full FGA framework achieves an out-of-distribution (OOD) average accuracy of 66.55%, representing a substantial improvement over existing approaches.
>
> | Algorithm | IN | IN-A | IN-V2 | IN-R | IN-Sketch | Avg. | OOD Avg. |
> |:----------|--:|---:|---:|---:|---:|---:|---:|
> | C-TPT+CoOp | 72.90 | 52.73 | 65.61 | 76.46 | 48.63 | 63.27 | 60.86 |
> | ZERO+CoOp  | 73.61 | 63.17 | 66.82 | 77.71 | 48.52 | 65.97 | 64.05 |
> | MTA+CoOp  | 73.99 | 59.29 | 66.97 | 78.20 | 49.96 | 65.68 | 63.61 |
> | SAPT+CoOp | 70.79 | 51.04 | 64.41 | 77.66 | 49.31 | 62.64 | 60.61 |
> | STSS+CoOp | 73.99 | 64.00 | 67.11 | 77.92 | 49.36 | 66.48 | 64.60 |
> | **FGA (Ours)** | **74.01** | **65.90** | **67.23** | **81.24** | **51.81** | **68.04** | **66.55** |
>
> In line with the established protocol from the seminal TPT paper, our current experiments focus on integrating recent TTA methods with CoOp. We acknowledge the potential of combining TTA with other prompt tuning approaches, and we plan to explore these directions in future extensions of this work. The latest results will be incorporated into Tables 1 and 2 later.
>
> **Q1-3:** **About Comparisons with Online TTA Methods**
>
> **A1-3:** Your valuable comments on adapting online TTA methods to the same experimental setting as ours are greatly appreciated. In response, we have adapted SAR into a similar test-time adaptation framework that utilizes sharpness-aware entropy minimization for prompt updates during inference. Notably, in the single-sample setting, SoTTA and SAR reduce to almost identical methods. The results presented below show that even the STSS component of FGA alone already outperforms SAR. Moreover, the full FGA method achieves an OOD average accuracy of 66.55%, which represents a substantial improvement over SAR's 61.75%.
>
> | Algorithm | IN | IN-A | IN-V2 | IN-R | IN-Sketch | Avg. | OOD Avg. |
> |:----------|--:|---:|---:|---:|---:|---:|---:|
> | SAR   | 73.03 | 55.35 | 65.89 | 77.09 | 48.65 | 64.00 | 61.75   |
> | SAPT+CoOp | 70.79 | 51.04 | 64.41 | 77.66 | 49.31 | 62.64 | 60.61 |
> | STSS+CoOp | 73.99 | 64.00 | 67.11 | 77.92 | 49.36 | 66.48 | 64.60 |
> | **FGA (Ours)** | **74.01** | **65.90** | **67.23** | **81.24** | **51.81** | **68.04** | **66.55** |
>
> We also appreciate your pointing out the issues regarding layout and typos. We will revise these in our final manuscript.

---

> > ### Comment · Reviewer_RFcj · 2025-11-26
> >
> > Thank you for the response. In my perspective, the additional results are crucial to strengthen the contribution. As ICLR allows to update the manuscript, could you please incorporate the suggestions/planned updates in the article?

---

> > > ### Author Response · Authors · 2025-11-30
> > >
> > > Thanks for your further comments and valuable suggestions. We have incorporated the additional results and further analysis into the latest version of our manuscript. Thanks again!

---

### Meta-Review · Area_Chair_QTzG · 2026-01-06

**Summary:**

Consider the four reviewers’ concerns on methodological coherence, experimental scope, theoretical rigor, and practical feasibility, the paper can be accepted.

**Reviewer Concerns:**

The authors addressed most concerns, e.g., Reviewer RFcj, Reviewer j4NZ, etc.

some minor concerns may need further inspectation, e.g., Independent validation of SAPT,  Quantitative training cost comparison, etc.

**Reviewer Scores:**

Reviewer RFcj may change the score for addressing his concerns.

---

### Decision · Program_Chairs · 2026-01-26

Accept (Poster)